# Assessment of Present and Future Water Security under Anthropogenic and Climate Changes Using WEAP Model in the Vilcanota-Urubamba Catchment, Cusco, Perú

Andrés Goyburo [1,*], Pedro Rau [2], Waldo Lavado-Casimiro [3], Wouter Buytaert [4], José Cuadros-Adriazola [4,5] and Daniel Horna [2]

1 Centro de Investigación y Tecnología del Agua (CITA), Universidad de Ingeniería y Tecnología (UTEC) and Dirección de Hidrología, Servicio Nacional de Meteorología e Hidrología (SENAMHI) and Programa de Maestría en Recursos Hídricos, Lima 15012, Peru
2 Centro de Investigación y Tecnología del Agua (CITA), Universidad de Ingeniería y Tecnología (UTEC), Lima 15063, Peru
3 Dirección de Hidrología, Servicio Nacional de Meteorología e Hidrología (SENAMHI), Lima 15072, Peru
4 Department of Civil and Environmental Engineering, Imperial College London, London SW7 2BX, UK
5 Consorcio para el Desarrollo Sostenible de la Ecorregión Andina (CONDESAN), Lima 15041, Peru
* Correspondence: agoyburo@senamhi.gob.pe

**Abstract:** Water is an essential resource for social and economic development. The availability of this resource is constantly threatened by the rapid increase in its demand. This research assesses current (2010–2016), short- (2017–2040), middle- (2041–2070), and long-term (2071–2099) levels of water security considering socio-economic and climate change scenarios using the Water Evaluation and Planning System (WEAP) in Vilcanota-Urubamba (VUB) catchment. The streamflow data of the Pisac hydrometric station were used to calibrate (1987–2006) and validate (2007–2016) the WEAP Model applied to the VUB region. The Nash Sutcliffe efficiency values were 0.60 and 0.84 for calibration and validation, respectively. Different scenarios were generated for socio-economic factors (population growth and increased irrigation efficiency) and the impact of climate change to evaluate their effect on the current water supply system. The results reveal that water availability is much higher than the current demand in the VUB for the period (2010–2016). For short-, middle- and long term, two scenarios were considered, "Scenario 1" (RCP 4.5) and "Scenario 2" (RCP 8.5). Climate change scenarios show that water availability will increase. However, this increase will not cover the future demands in all the sub-basins because water availability is not evenly distributed in all of the VUB. In both scenarios, an unmet demand was detected from 2050. For the period 2071–2099, an unmet demand of 477 hm$^3$/year for "Scenario 1" and 446 hm$^3$/year for "Scenario 2" were estimated. Because population and agricultural demands are the highest, the effects of reducing the growth rate and improving the irrigation structure were simulated. Therefore, two more scenarios were generated "Scenario 3" (RCP 4.5 with management) and "Scenario 4" (RCP 8.5 with management). This socio-economic management proved to be effective in reducing the unmet demand up to 50% in all sub-basins for the period 2071–2099.

**Keywords:** WEAP model; water balance; water security; climate change; Andean basin; hydrological modeling

## 1. Introduction

Water resources are essential for improving the quality of life, economic development, and ecological stability [1]. To measure this improvement, water security is assessed. Water security is the availability of an acceptable quantity and quality of water for health, livelihoods, ecosystems, and production, along with an acceptable level of water-related risks to people, the environment, and the economy [2]. Ensured water security has been

increasingly challenged due to the growth of multiple factors, such as climate change, population expansion, groundwater depletion, increased energy demand, and environmental flow requirements [3]. This growth gives rise to a scarcity of water, a condition in which water demand grows beyond available water supply [4]. The Intergovernmental Panel on Climate Change (IPCC) has identified the following two significant signs of climate change: global temperature increases due to greenhouse gas emissions and water cycle changes [5]. Climate and water are intricately connected through several variables such as precipitation, temperature, and solar radiation, which differ from one region to another [6]. In the tropical Andes, there is a complex interaction between the hydroclimate and the accelerated melting of ice caused by climate change, which has a significant impact on socioeconomic activities. Hence, this challenges local and regional livelihoods and water resource management [7].

Peru has a unique combination of rainfall variability and landscapes that generate different hydrologic conditions and water availability [8]. The capacity for cities to cope with climate change is further stressed by socioeconomic drivers such as population growth and water consumption intensity [9,10]. Although previous work has included climate change scenarios for future water security in the Andes [11–13], few of them have included socioeconomic drivers to assess if local adaptation strategies would be enough to guarantee future water demand satisfaction [14]. A potential reason for the inexistence of this exercise is data scarcity, both in terms of hydrometeorological and socioeconomic observations in the Andes. This has hindered the possibility of making accurate predictions with complex hydrological models and has instead favored the use of simpler conceptual models [13].

This study examines the potential of the currently proposed adaptation strategies in the VUB region to offset the impacts of climate change on the hydrological cycle and the rising water demand at the current increasing rates. This is performed by combining the WEAP model with the best available hydrometeorological datasets, the downscaled climate change scenarios, and socioeconomic data from governmental official sources in the watershed. The WEAP model has been previously used in Peru for climate change studies, for instance; one such study evaluated the impact of climate change on mountain hydrology [15] and another on improvement of water use for irrigation [16]. For both case studies the model was adequate, even in catchments with few observations available. In addition, the WEAP model is widely used to assess water security in a watershed under climate change and/or socio-economic changes [17]. Because the system is designed to examine alternative water management strategies based on the water balance principle, by considering demand priorities and supply preferences, it provides a set of entities and procedures to explore and find solutions to the problems decision-makers face using a scenario-based approach [18].

## 2. Materials and Methods

### 2.1. Study Area

The study area is located in southern Peru, in the department of Cusco. It covers the upper part of the Urubamba basin. The study area is called Vilcanota-Urubamba since the main tributary is the Vilcanota River in the highest part. The middle part is called the Urubamba River, up to the Pisac hydrometric station. Hereinafter, the study area will be referred to as "VUB". Figure 1 shows the study area, which has an area of 6903 km$^2$ and an altitude that ranges between 3000 and 6000 m.a.s.l.

Precipitation is highly seasonal, with the wet season from November to March and the dry season from June to August. During the wet season precipitation is between 120 and 150 mm and during the dry season it is between 70 and 0 mm. Therefore, the mean precipitation is 797 mm. Regarding the temperature, the minimum ranges from −5 °C to 4 °C, while the maximum from 16 °C to 19 °C.

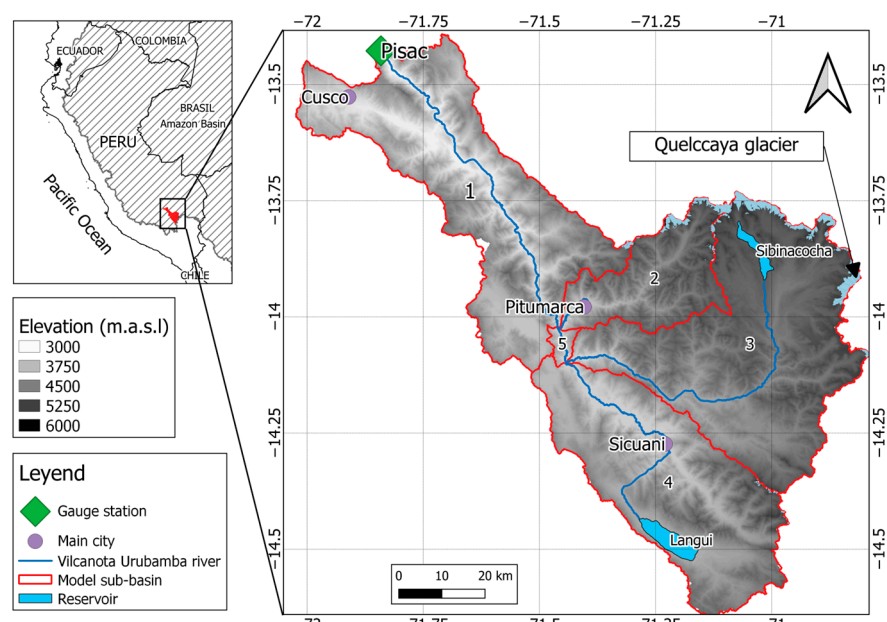

**Figure 1.** Study area showing the location of the hydrometric station, towns, rivers, and reservoirs.

Figure 2 shows that the most common land cover is the high Andean grassland (56.9%), followed by bare soil (12.8%), scrub (11.4%), and agriculture (9.8%). The watershed has a broad upper part due to the intense erosion of the glacial and alluvial origin. The lower part of the basin is narrower due to its lithology, which corresponds mainly to hard sedimentary rocks that do not allow lateral erosion processes, with vertical erosion predominating.

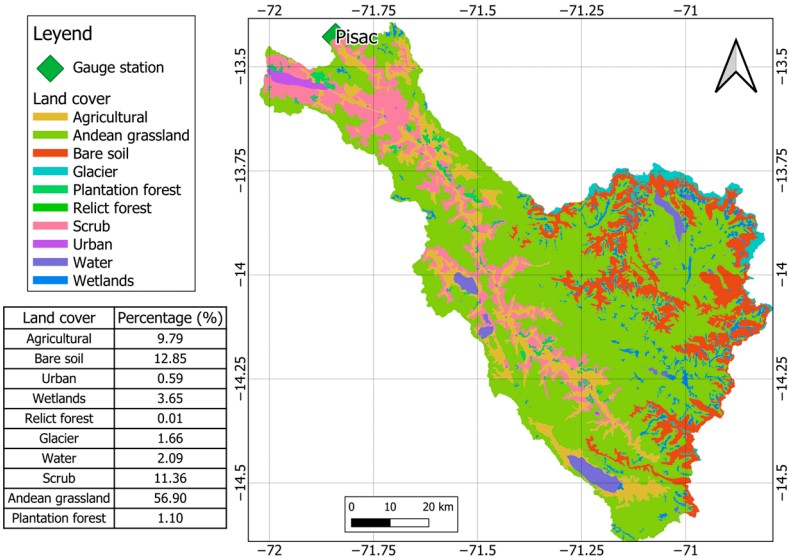

**Figure 2.** The land cover/use map of the VUB region is derived from the National Ecosystem Map Data [19].

The study area presents diverse hydrogeology although fractured aquifers are the most representative features. These aquifers are characterized by quartzite, sandstone, shale, and silt clay. There is a flow between fractured and deep flows. The type of aquifer is of the fissured sedimentary type [20].

The study area was delimited into five sub-basins to represent different physiographic characteristics to be used in the hydrological model, considering the hydrometric station "Pisac" in sub-basin 1 as the outflow point. Details of the main physiographic characteristics of each sub-basin are shown in Table 1.

**Table 1.** Physiographic characteristics of sub-basins.

| Sub-Basin No | Area (km$^2$) | Minimum Elevation (m.a.s.l) | Mean Elevation (m.a.s.l) | Maximum Elevation (m.a.s.l) | Average Slope in Degrees (°) | Main Channel Length (km) |
|---|---|---|---|---|---|---|
| 1 | 2099 | 2998 | 3867 | 4736 | 17 | 60 |
| 2 | 684 | 3567 | 4632 | 5697 | 19 | 16 |
| 3 | 2333 | 3461 | 4610 | 5759 | 15 | 64 |
| 4 | 1742 | 3481 | 4166 | 4851 | 12 | 39 |
| 5 | 42 | 3430 | 3950 | 4470 | 16 | 9 |

*2.2. Data Collection*

Different data sources were used to develop the hydrological model (Table 2). The hydrologic data used in the calculation are precipitation, evapotranspiration, and streamflow. The climate data were obtained from the gridded products PISCO ("Peruvian Interpolate Data of the SENAMHI's Climatological and Hydrological Observations") [21,22] and meteorological stations. The GCMs come from the NEX-GDDP [23], consisting of 21 models for the RCP scenarios 4.5 and 8.5. The daily streamflow data record was collected between 1987 and 2016 from the hydrometric station PISAC, which belongs to SENAMHI (National Meteorological and Hydrological Service) and is located at the outflow of the entire VUB region. The vegetation cover was obtained from the "National Ecosystem Map" produced by MINAM (Ministry of Environment) [19], which is the most up-to-date study available for the study area. Population statistics were obtained from the Peruvian National Census, carried out by the INEI (National Institute of Statistics and Informatics) in 2017. Water demand data were obtained from the National Water Authority (ANA)-Urubamba Vilcanota (AAA-UV) through the Administrative Registry of Water Use Rights (RADA in Spanish).

**Table 2.** Sources for the data used.

| Data | Description | Source |
|---|---|---|
| Climate | Precipitation, temperature, humidity, wind speed | PISCO [21] SENAMHI (PERU) |
| Climate change | NEX-GDDP: Downscaled Climate Projections (2017–2099) 21 GCMs: RCP scenarios 4.5 and 8.5 | NEX-GDDP ee.ImageCollection("NASA/NEX-GDDP") * |
| Remote sensing | Digital Elevation Model 30 m: SRTM Land cover 1.5 m "National Ecosystem Map" | ee.Image("USGS/SRTMGL1_003") * MINAM (Perú) [19] |
| Hydrometric | Hydrometric station Pisac (1987–2016) | SENAMHI PERU |
| Water demand | Population "Population water use licenses" | INEI (Perú) AAA-UV (Perú) |

Note: * ee.ImageCollection() is a function in Google Earth Engine to access an imaginary database.

*2.3. Methodology*

This study focuses on developing an implementation of the WEAP model and its calibration and validation for five sub-basins in the VUB. Following the development of the model, four scenarios are used to predict water security in the short- (2017–2040), middle- (2041–2070), and long term (2071–2099). The WEAP model was run monthly. The period 2010–2016 was used as the base period in the model for estimating population, agricultural, industrial, and energy demands. Future demands were projected up to 2099 and averaged as short-, middle-, and long term, keeping constant the baseline population growth for population demand, future climate data for agricultural demand, the industrial growth trend, and the energy demand. The climate change scenarios used were from the global climate models (RCP 4.5 and RCP 8.5) to estimate the water balance up to 2099 and averaged as short-, middle-, and long term. Figure 3 displays the working scheme and the data used to implement the WEAP model.

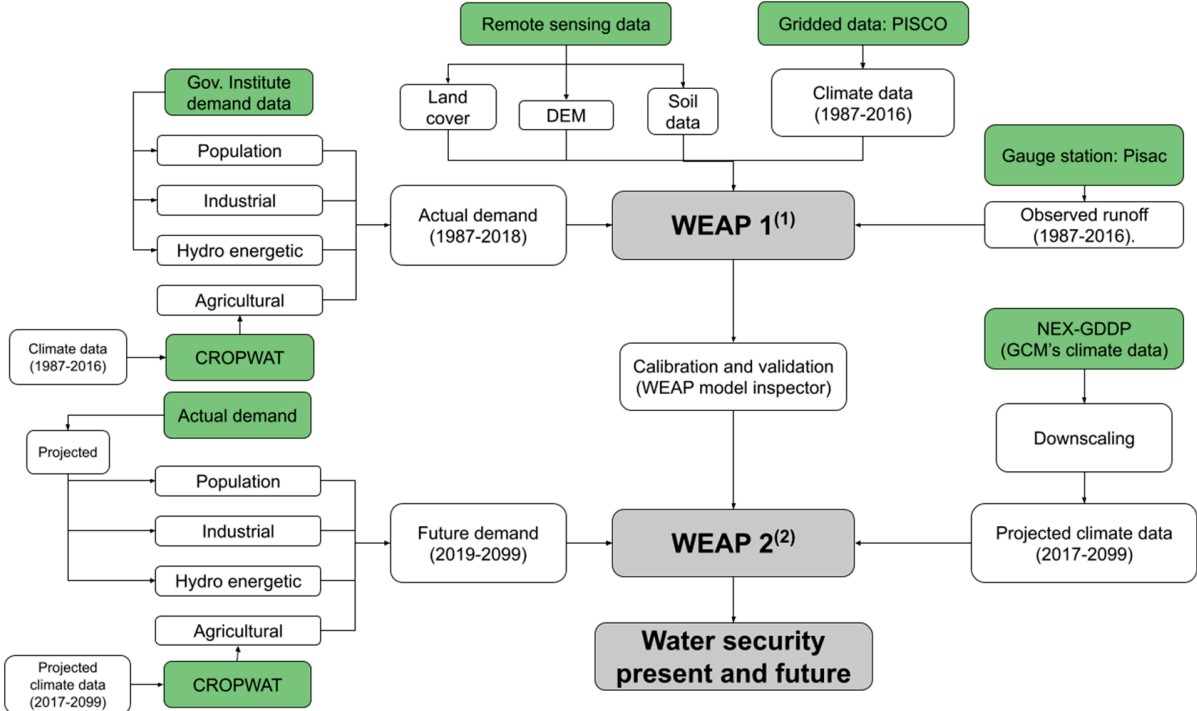

**Figure 3.** Methodological scheme of the research. [1] Represents the WEAP model before the calibration and validation process and [2] Represents the calibrated and validated WEAP model where the projected data are used to estimate present and future water security.

The model runs until the year 2099. Starting conditions for water supply and demand were set for 2016. Calibration was carried out from 1987 to 2006, and validations from 2007 to 2016. During this period, four different scenarios were evaluated (Table 3). The maximum population growth in the sub-basins is 1.8% annually, which could be reduced to 0.3% by applying migration control policies [24]. An irrigation efficiency of 50% represents the lack of technical irrigation. In the future, the use of technical irrigation is expected to increase with government funding [25], increasing irrigation efficiency to 80%.

**Table 3.** Characteristics of the scenarios considered for this study.

| Name | GCM Scenario | Socio-Economic Factors | |
| --- | --- | --- | --- |
| | | Population Growth (%) | Irrigation Efficiency (%) |
| Scenario 1 | RCP 4.5 | 1.8 | 50 |
| Scenario 2 | RCP 8.5 | 1.8 | 50 |
| Scenario 3 | RCP 4.5 | 0.3 | 80 |
| Scenario 4 | RCP 8.5 | 0.3 | 80 |

Model Evaluation Statistics

In the sensitivity analysis, two statistics were used to quantify how well the simulated data represent the observed data. The first is the Nash–Sutcliffe coefficient of efficiency (NSE) Equation (1) and the second is the percentage deviation (PBIAS) Equation (2). Table 4 shows the ranges of values used for the classification of the metrics used in each run according to [26].

$$\text{NSE} = 1 - \frac{\sum_{i=1}^{n} \left( Q_i^o - Q_i^s \right)^2}{\sum_{i=1}^{n} \left( Q_i^o - Q_i^m \right)^2} \tag{1}$$

$$\text{PBIAS} = \frac{\sum_{i=1}^{n} \left( Q_i^o - Q_i^s \right) \times 100}{\sum_{i=1}^{n} Q_i^o} \tag{2}$$

where $Q_i^o$ is the *i*th flow observation, $Q_i^s$ is the ith flow simulated, $Q_i^m$ is the mean of observed flow data, and n is the total number of observations.

**Table 4.** General performance ratings for statistics [26].

| Performance Rating | PBIAS (%) | NSE |
|---|---|---|
| Very good | PBIAS < ±10 | 0.75 < NSE ≤ 1.00 |
| Good | ±10 ≤ PBIAS < ±15 | 0.65 < NSE ≤ 0.75 |
| Satisfactory | ±15 ≤ PBIAS < ±25 | 0.50 < NSE ≤ 0.65 |
| Unsatisfactory | PBIAS ≥ ±25 | NSE ≤ 0.50 |

*2.4. Hydrological Model*

WEAP ("Water Evaluation and Planning") is a water allocation model that provides a comprehensive approach to water resource planning [27]. The model relies on the water balance to replicate the behavior of the hydrological cycle [28]. It estimates water resources by integrating hydrology, land use, hydrogeology, climate, water quality, and anthropogenic effects [29]. WEAP is a conceptual model, thus allowing the representation of physical systems [30]. Watersheds, rivers, lagoons, and hydraulic structures such as reservoirs and hydropower plants can be simulated using additional components.

There are five different methods for the simulation of watersheds in WEAP [27]. This research employed the "Soil Moisture" method for estimating rainfall runoff, which has been widely used [31–36]. This methodology divides the soil into two layers (Figure 4). The upper soil layer is called Bucket 1 and simulates runoff, sub-surface flow, evapotranspiration, and soil moisture. The lower soil layer called Bucket 2 simulates percolation and baseflow, which may be conveyed to an aquifer or a river. For a basin divided into many sub-basins by land use or type, the water balance is calculated for each area j for the first layer, assuming that the climate is constant for each sub-basin [6]. The equation for the water balance is shown below: [27].

$$\mathrm{Rd}_j \frac{d Z_{1,j}}{dt} = P_e(t) - \mathrm{PET}(t) k_{c,j}(t) \left( \frac{5 Z_{1,j} - 2 Z_{1,j}^2}{3} \right) - P_e(t) Z_{1,j}^{\mathrm{RRF}_j} - f_j k_{s,j} Z_{1,j}^2 - (1 - f_j) k_{s,j} Z_{1,j}^2 \tag{3}$$

where $Z_{1,j}$ is the relative storage, referring to the total effective storage in the root zone, and $\mathrm{Rd}_j$ is the soil retention capacity of the soil cover in area j (mm). PET is calculated using the Penman–Monteith method using the crop or plant coefficient ($k_{c,j}$) for each covered area. $P_e$ is the effective precipitation, and RRFj is the runoff resistance coefficient corresponding to the area covered. $P_e(t) Z_{1,j}^{\mathrm{RRF}_j}$ is the surface runoff; $f_j\, k_{(s,j)} Z_{1,j}^2$ is the inflow to the first layer; the term $k_{s,j}$ is the conductivity in the saturated root zone (mm/time); $f_j$ is the coefficient dividing horizontal and vertical flow depending on soil, cover, and topography.

*2.5. The Hydrological Scheme in WEAP*

Once the data had been collected and analyzed, the WEAP model was used to determine the water balance. Figure 5 illustrates the schematic of the VUB region, ranging from 2010 to 2099. The starting condition year is 2010, and the simulation was conducted from 2011–2099. The calibration period for the observed data is from 2010 to 2016.

The catchment simulation method is the Rainfall-Runoff method (Soil Moisture). This method is more complex, representing a catchment with two soil layers. For this method, we characterize the land cover (Figure 2) with every parameter described in Equation (1).

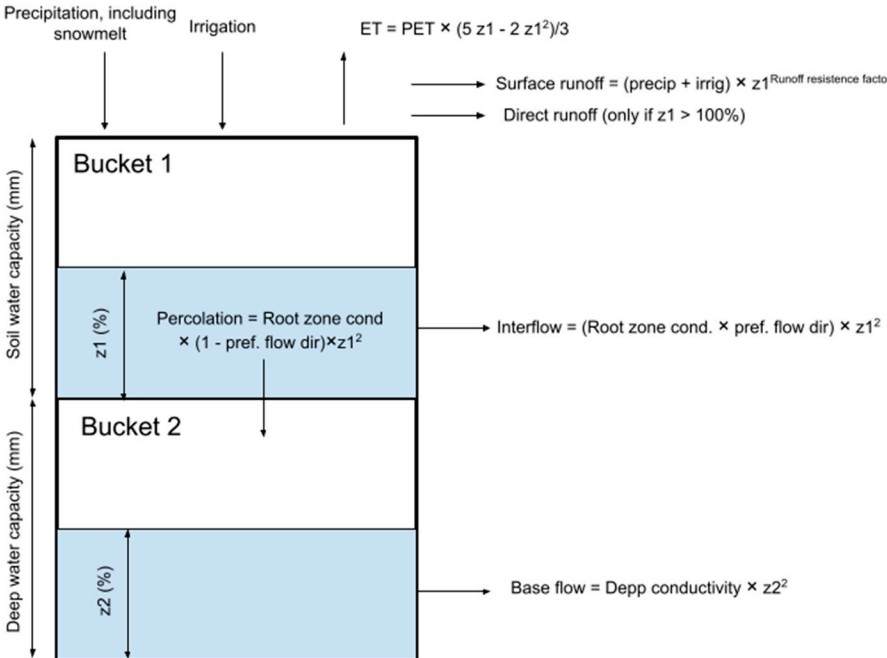

**Figure 4.** Conceptual scheme and equations of the soil moisture method [27].

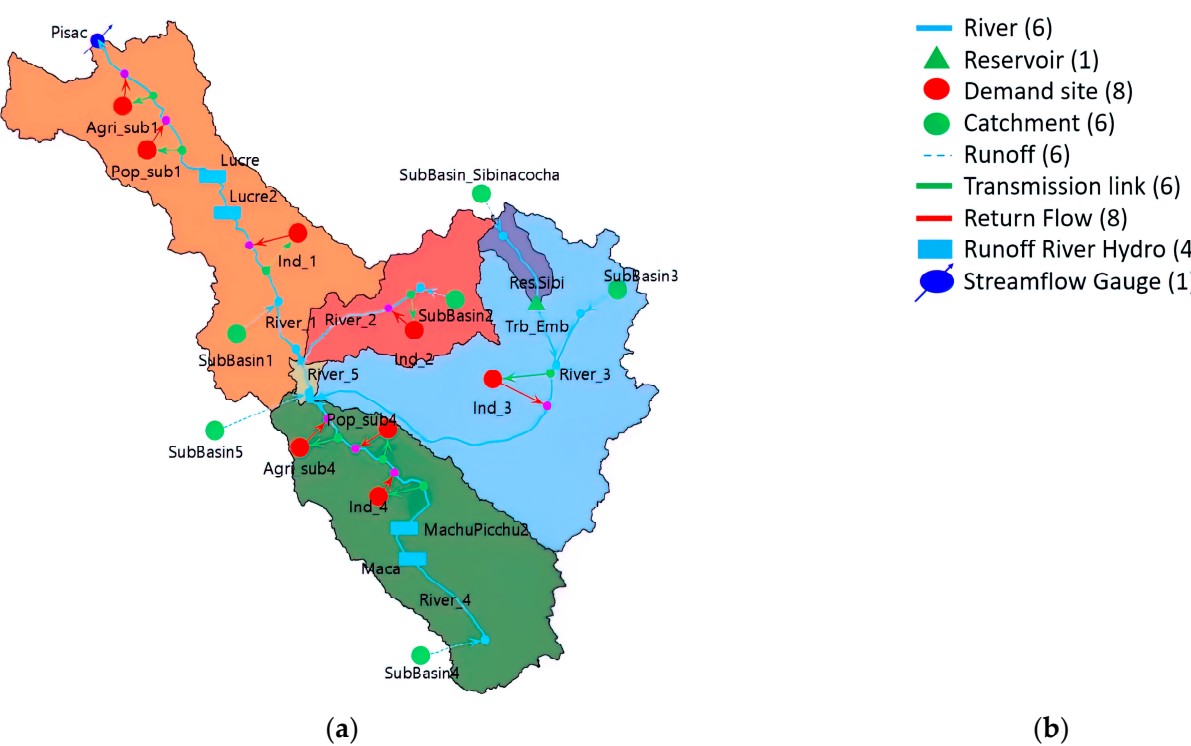

(**a**)                                                                                    (**b**)

**Figure 5.** (**a**) Schematic of the VUB region in the WEAP model. (**b**) Legend.

Key assumptions for future modeling:

- Vegetation cover remains constant until 2099.
- The agricultural area remains constant until 2099.
- Sibinacocha reservoir operates until 2099 with its current operating rules.
- Run-of-river power plants with their operating flows will meet the energy demands until 2099.
- Snow cover and aquifers were not included because there is insufficient data for modeling.

This study focuses on population growth and climate change because these are the biggest pressure variables in the hydrological cycle in the Andes [20]. We do not consider land use and land cover changes because it requires additional socioeconomic data as proxy variables, and our study area exhibits a lack of information. This can lead to considerable uncertainty in projections.

*2.6. Current and Future Water Demands*

The following four main types of demand were identified: population, agricultural, industrial, and energy. These demands were estimated on an annual scale and assigned to each sub-basin.

- Population: This was estimated using the "population water use licenses" in the corresponding sub-basins and the number of inhabitants in the districts within the sub-basins. The population growth rate was projected using Equation (4), linear increase/decrease has been steady over the last 13 years in Cusco according to INEI [24]. The relationship between water demands and the number of inhabitants can be used to reconstruct the past demand and project the demand up to the year 2099:

$$r = \frac{\sqrt[t]{P_t}}{P_o} - 1 \qquad (4)$$

  where r is growth rates, $P_t$ is the final population, T is the number of years, and $P_o$ is the initial population.

- Agricultural: Since there is no reliable data on demand for the water use licenses, estimation was performed using Equation (5). CROPWAT software was used to estimate the crop water requirement, and the irrigation efficiency is equal to 0.5, according to [25]. The agricultural area for each sub-basin was obtained from the ecosystem map [19] to maximize the demands; it was considered that all the agricultural area is in use and will remain constant until 2099. Overestimations in water demand were proposed as there are no measurements of actual water use. Cropping census data were obtained from the study conducted by the ANA (2015). Climate variables obtained from the GCMs were used for future demand:

$$DA = \frac{RHC}{Efc} \times A, \qquad (5)$$

  where DA is agricultural demand ($m^3$/year), RHC is crop water requirement, Efc is irrigation efficiency, and A is the area of the crop

- Industrial: These licenses were found in the RADA—industrial, recreational, mining, aquaculture, and other uses. These types of uses were converted to industrial uses because they are economic activities. As with the previous demands, they were grouped according to their location in each sub-basin.
- Energy: This demand corresponds to the amount of water granted to energy-generating companies for their operations. The primary source of water for generation is the Sibinacocha reservoir. Table 5 shows the operating characteristics of the reservoir.

**Table 5.** Reservoir Characteristics.

| Description | Volume ($hm^3$) |
|---|---|
| Storage capacity | 120 |
| Useful volume | 110 |
| Bottle dead | 10 |

*2.7. Climate Change Projections*

The average multi-model ensemble process was performed to reduce the inherent uncertainty of each GCM model [37]. According to [38], a multi-model ensemble has better results than using each model individually. The multi-model ensemble results from

averaging all the GCM models downloaded for each precipitation and temperature variable. The models used were as follows: ACCESS1-0, bcc-csm1-1, BNU-ESM, CanESM2, CCSM4, CESM1-BGC, CNRMCM5, CSIRO-Mk3-6-0, GFDL-CM3, GFDL-ESM2G, GFDL-ESM2M, INMCM4, IPSLCM5A-LR, IPSL-CM5A-MR, MIROC-ESM, MIROC-ESM-CHEM, MIROC5, MPIESM-LR, MPI-ESM-MR, MRI-CGCM3, NorESM1-M.

It is impossible to use GCMs directly at a smaller scale due to the large scale of the spatial resolution [39]. Therefore, a statistical downscaling procedure was used to address this limitation. In this case, the non-parametric process called "quantile mapping" [40] was used. This method was applied because it adjusts the mean of the values and the shape of the distribution without modifying the chronological structure of the time series [31]. Moreover, it has been widely used by [11,31,41,42]. Climate data from the PISCO precipitation and temperature products were used for downscaling purposes as the observed data for the period 1981–2016, with an adjustment of the average ensemble data for 2017–2099.

### 2.8. Quantile Mapping Procedure

Due to the large scale of the spatial resolution of GCMs, it is not possible to use them directly on a smaller scale [39]. To overcome this problem, the statistical downscaling procedure was used. The quantile mapping process is detailed by [42] as follows:

Quantile mapping (also called quantile matching, cumulative distribution function matching, or quantile-quantile transformation) attempts to find the transform of the modeled variable $P_m$ such that the new distribution is equal to the distribution of the dependent variable $P_o$. This transformation is generally expressed as Equation (6):

$$P_o = \mathrm{h}(P_m) \tag{6}$$

Statistical transformations are an application of the probability integral transform [43] and if the distribution of the variable of interest is known, the transformation is defined as Equation (7):

$$P_o = F_o^{-1}(F_m(P_m)) \tag{7}$$

where $F_m$ is the CDF of $P_m$ and $F_o^{-1}$ is the inverse CDF (or o quantile function) corresponding to $P_o$.

## 3. Results

### 3.1. Model Calibration and Validation

During the calibration period (1987–2006), the model underestimated (30%) the flow during the wet season, but in the validation period (2007–2016), the model overestimated (5%) the flow during the wet season (Figure 6). The simulated flows in the dry water season, on the other hand, represent well (error less than 5%) in the calibration period but are overestimated (10%) in the validation period.

Notwithstanding, the model produces reasonable values for NSE and PBIAS (Table 6). The NSE values for the calibration and validation periods are 0.60 and 0.84, respectively. The PBIAS is 12.8% for the calibration period and 8.5% for the validation period. As per [26], the NSE values are classified as "very good" for the calibration period and "good" for the validation period; the PBIAS values are classified as "very good" for the calibration period and "good" for the validation period.

Hence, this demonstrates that the model can simulate the hydrology of the study area on an accurate basis. Consequently, it offers a high degree of confidence that the prediction of future flows will be accurate by maintaining the calibrated parameters up to the year 2099. This methodology has been used in multiple research studies: [3,29,31,32,44,45].

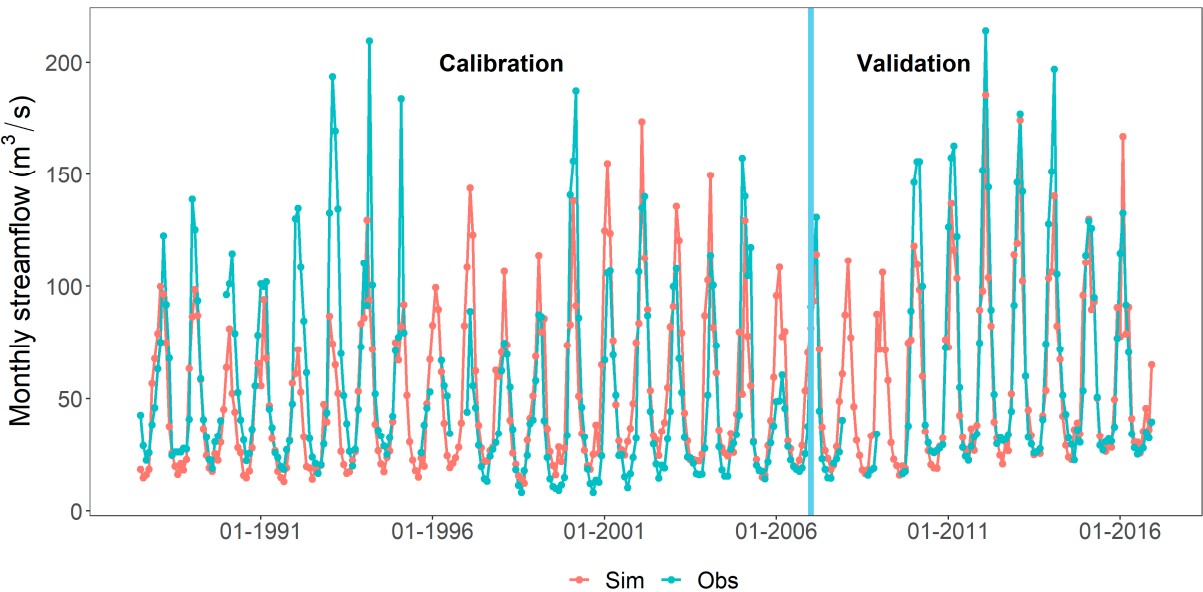

**Figure 6.** Monthly average of simulated and observed flow (1987–2016).

**Table 6.** Values of simulated (Qs) and observed (Qo) average flow metrics.

| Simulation | NSE | PBIAS (%) | Qo (m³/s) | Qs (m³/s) |
|---|---|---|---|---|
| Calibration (1987–2006) | 0.60 | 12.8 | 52.3 | 59 |
| Validation (2007–2016) | 0.84 | 8.5 | 64.2 | 69.7 |

*3.2. Initial Demand and Water Security*

The current estimated demand is shown in Table 7. The estimated total demand is 254 hm³/year (2010). Agricultural demand is the largest, with 220 hm³/year, equivalent to 86.6%. Population use demands 30.3 hm³/year, representing 12%. Industrial use is the lowest, with 3.6 hm³/year representing only 1.4%. The energy demand is 266.6 hm³/year, but since it is a "non-consumptive" use, it was not considered in the total demand. Further description can be found in [46].

**Table 7.** Actual demand in the VUB region.

| Demand | Volume (hm³) | Percentage (%) |
|---|---|---|
| Population | 30.3 | 12 |
| Agricultural | 220 | 86.6 |
| Industrial | 3.6 | 1.4 |

Water Security in the Period 2010–2016

According to the analysis of water availability, there is a positive balance between water supply and demand. The model shows a yearly runoff of 2237 hm³/year from the Pisac station. Figure 7 indicates the amount of water entering and draining out of the study area. A significantly higher amount of water enters during the wet season at 572 hm³ than during the dry season at 42 hm³. Water outflow is higher during the wet season at 443 hm³ and in the dry season at 302 hm³. Water outflow is higher during the dry season because the main source of inflow, precipitation, decreases and all the water in the soil is released.

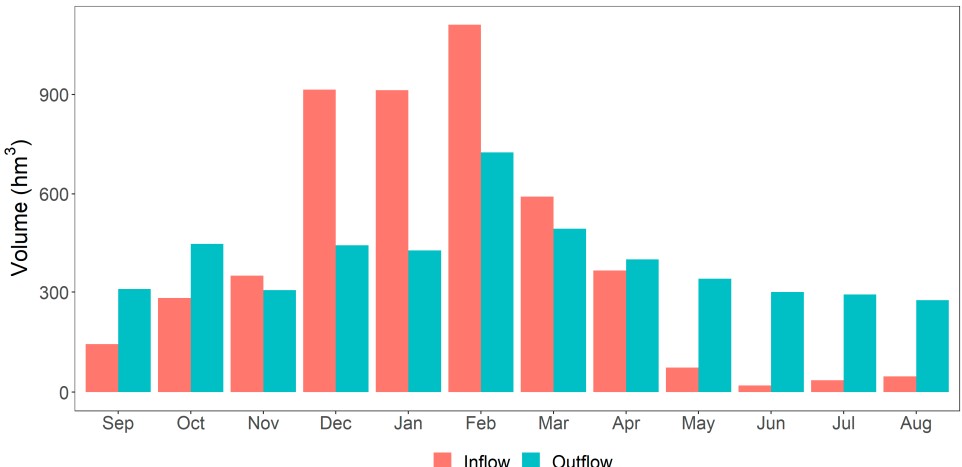

**Figure 7.** Monthly average of water inflows and outflows to the study area (2010–2018).

There is a considerable surplus of water to be used. When comparing the amount of precipitation and runoff with the demands, they are respectively 19 and 8 times higher. This is only 5.3% in terms of precipitation and 13% in terms of runoff. Therefore, this surplus would allow demands to increase or intensify, e.g., by 10 to 15 times. However, availability is not equally distributed in the VUB. The highest availability is found in the Vilcanota River in Sub-basin 1, as it is located at a lower altitude and receives runoff from the other sub-basins.

### 3.3. Future Water Demand and Security

Future demands were projected up to the year 2099, based on their behavior in the period 2010–2018. Then, every type of demand was averaged in short- (2017–2040), middle- (2041–2070), and long-term (2071–2099). Each one is explained below.

#### 3.3.1. Population Demand

Approximately 99% of the entire population of the basin is concentrated in Sub-basins 1 and 4, therefore, only these were taken into account for the analysis. According to the water use licenses, a second-degree polynomial relationship was determined between the number of inhabitants and the annual demand. The population in Sub-basin 1 is estimated to grow at a rate of 1.8% and in Sub-basin 4 at a rate of 0.2% using Equation (4). There is a high migration of inhabitants from higher altitudes (Sub-basin 4) to lower altitudes (Sub-basin 1), which is reflected in their growth rates. Figure 8 presents the relationships obtained for each sub-basin, and these equations were used to project the population's water demand until 2099.

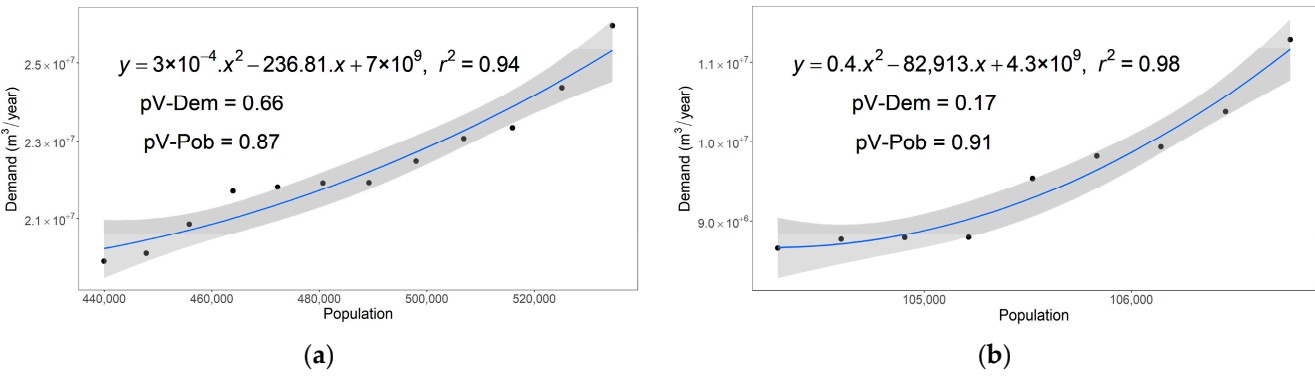

**Figure 8.** Relationship between water demand and inhabitants, (**a**) Sub-basin 1 and (**b**) Sub-basin 4.

The highest growth in demand occurs in Sub-basin 1 (Figure 9). In addition, the estimated population for the year 2099 is 2.2 million with a demand of 1047 hm$^3$ for Sub-basin 1 and 135,375 inhabitants with 393 hm$^3$ of demand in Sub-basin 4 as show in Table 8.

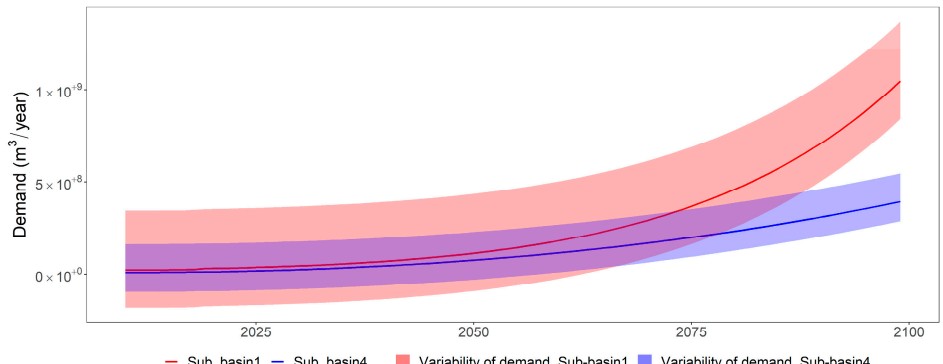

**Figure 9.** Projected population demand (2010–2099).

**Table 8.** Annual population demand in the VUB region.

| Year | Sub-Basin 1 | | Sub-Basin 4 | |
|---|---|---|---|---|
| | Population | Demand (hm$^3$) | Population | Demand (hm$^3$) |
| 2030 | 661,032 | 44.5 | 110,589 | 24.2 |
| 2040 | 789,051 | 69.9 | 113,878 | 45.1 |
| 2050 | 941,863 | 113.1 | 117,265 | 75.5 |
| 2060 | 1,124,270 | 182.9 | 120,753 | 116.4 |
| 2070 | 1,342,002 | 292.5 | 124,345 | 168.6 |
| 2080 | 1,601,901 | 460.5 | 128,043 | 233.1 |
| 2090 | 1,912,134 | 714.1 | 131,851 | 310.9 |
| 2099 | 2,242,398 | 1047.4 | 135,375 | 393.1 |

The estimated average per capita water consumption for the period 2030–2099 was estimated at 743.6 L/inhabit/day (Sub-basin 1) and 3650.4 L/inhabit/day (Sub-basin 4), thus overestimating the values determined by [47] of 100 L/inhabit/day. This overestimation is due to the second-degree polynomial regression used; if a linear regression were to be used, it would represent future values more adequately. In contrast, according to [48] the consumption per person in the Highlands is 1000 L/inhabit/day. On this basis, the value obtained in Sub-basin 4 would be considered an overestimation. In Sub-basin 4, it may be the case that licenses for population use are also being used for major economic activities, increasing population water consumption.

### 3.3.2. Agricultural Demand

As shown in Figure 10, the demands decrease up to the year 2099. By keeping the efficiency constant (0.5), the crop water requirement decreases. This reduction occurs because precipitation and temperature increase in all scenarios according to the multi-model ensemble. In Sub-basin 1, precipitation will increase by 1.1% by 2099 for both scenarios, whereas in Sub-basin 4, it will increase on average by 1.5%. By increasing precipitation, the current crop requirement decreases. According to Table 9, the agricultural demand in Sub-basin 1 for 2099 will be 106 hm$^3$ and 111 hm$^3$, with a maximum of 154.8 hm$^3$ and 157.9 hm$^3$ for Scenario 1 and Scenario 2. In Sub-basin 4 for 2099, it will be 34 hm$^3$ and 27 hm$^3$, with a maximum of 66.7 hm$^3$ and 58.3 hm$^3$ for Scenario 1 (RCP 4.5) and Scenario 2 (RCP 8.5).

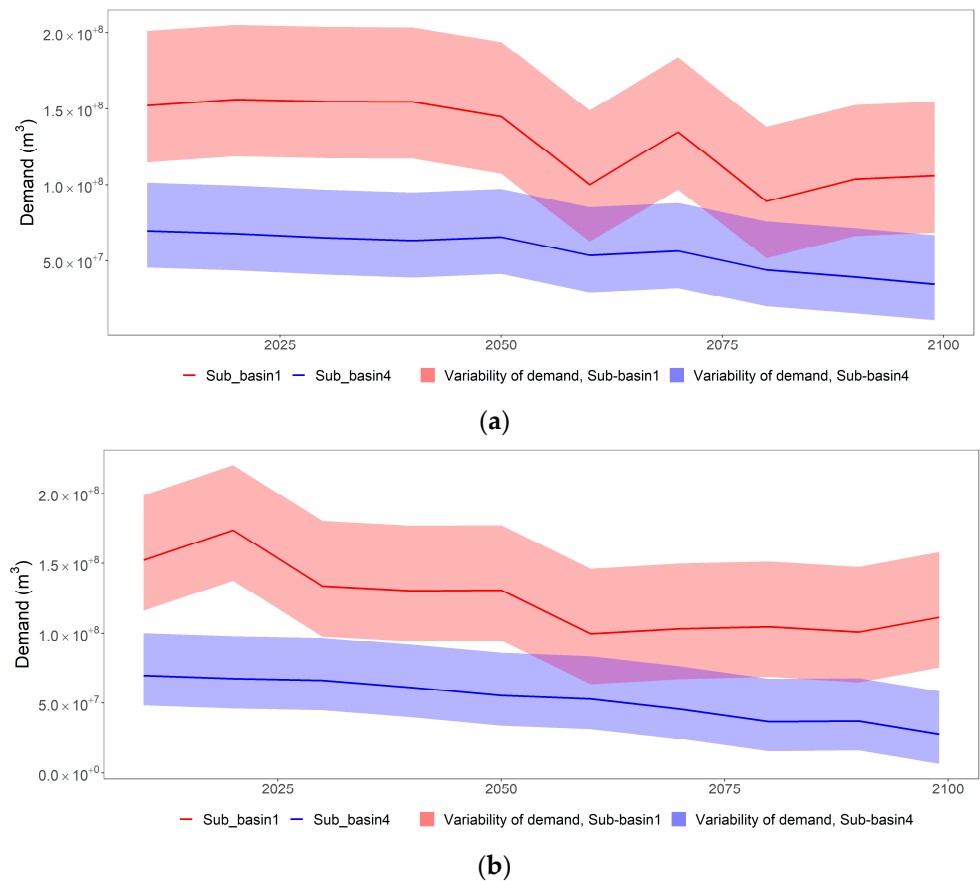

**Figure 10.** Estimated agricultural demands for (**a**) Scenario 1 and (**b**) Scenario 2 (2020–2099).

**Table 9.** Future agricultural demand.

| Year | Sub-Basin 1 (hm³) | | Sub-Basin 4 (hm³) | |
|---|---|---|---|---|
| | Scenario 1 (RCP 4.5) | Scenario 2 (RCP 8.5) | Scenario 1 (RCP 4.5) | Scenario 2 RCP (8.5) |
| 2030 | 156.2 | 173.5 | 67.7 | 67.3 |
| 2040 | 154.9 | 133.7 | 64.9 | 66.1 |
| 2050 | 154.5 | 130.3 | 63.1 | 61.0 |
| 2060 | 144.8 | 130.6 | 65.4 | 55.2 |
| 2070 | 100.1 | 99.6 | 53.3 | 52.7 |
| 2080 | 134.5 | 103.3 | 56.9 | 45.5 |
| 2090 | 891.9 | 104.7 | 44.1 | 36.3 |
| 2099 | 103.7 | 100.8 | 39.5 | 37.2 |

### 3.3.3. Industrial Demand

According to the water use licenses, it was determined that there is a linear trend in the growth of industrial demands. In Figure 11, Sub-basin 1 has higher growth than Sub-basin 4. Indeed, this is because the largest population is concentrated in Sub-basin 1. The projected demand in 2099 is 31 hm³ for Sub-basin 1 and 5 hm³ for Sub-basin 4 (Figure 12).

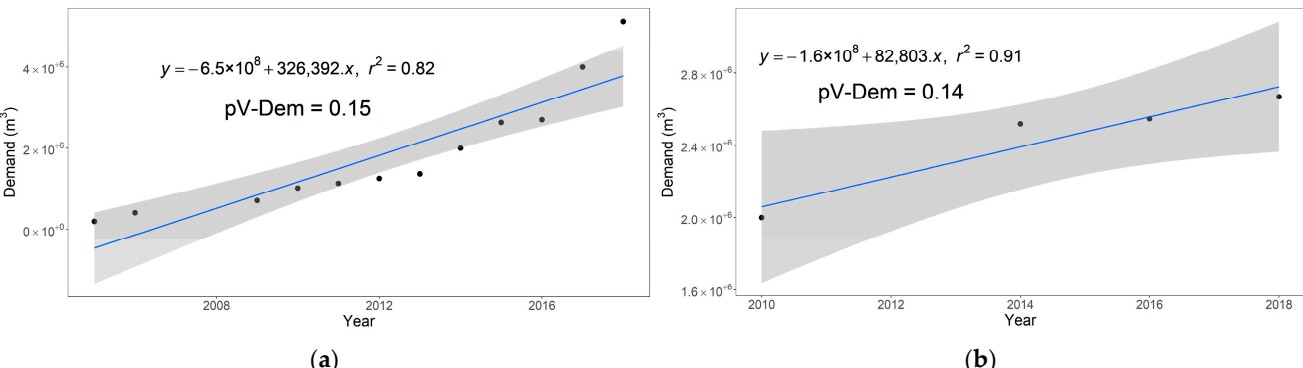

**Figure 11.** The positive linear trend of industrial demands, (**a**) Sub-basin 1, (**b**) Sub-basin 4.

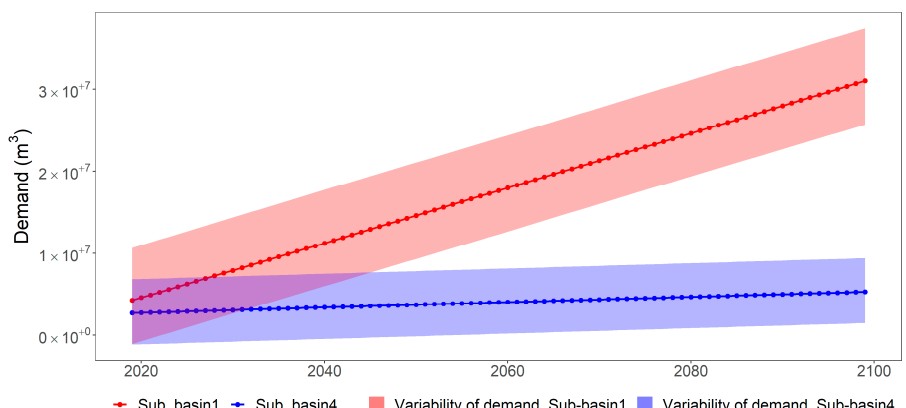

**Figure 12.** Estimated industrial demand Scenario 1 and Scenario 2 (2020–2099).

### 3.4. Future Scenario in the Context of Climate Change (CC)

Figure 13 shows that for the short term (2017–2040), the water supply will cover all demands, so there is no unmet demand in either scenario. In the middle term (2041–2070), unmet demand is estimated at 56.72 hm$^3$ for Scenario 1 and 42.62 hm$^3$ for Scenario 2. For the long term (2071–2099), the unmet demand is 477.21 hm$^3$ in Scenario 1 and 445.80 hm$^3$ in Scenario 2. The period 2071–2099 is the most critical since the unmet demand increases 8–10 times compared to the previous period. In both scenarios, Sub-basins 1 and 4 have the highest volumes of unmet demand. The main types of unmet demand for both sub-basins are agricultural and population-related issues. Population demand is constantly increasing due to the growth rates in both catchments.

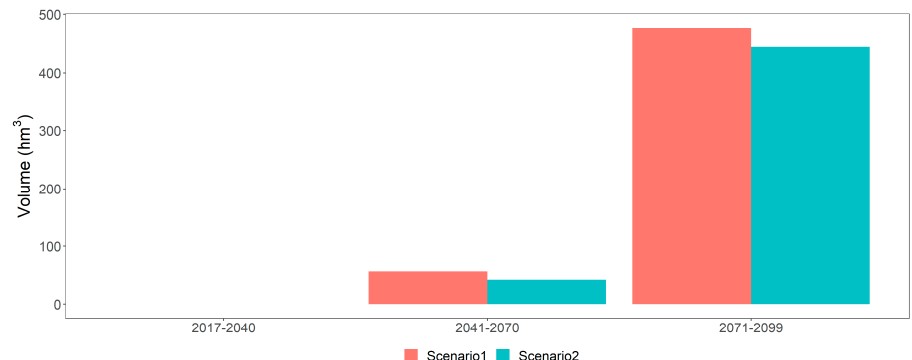

**Figure 13.** Annual unmet demand for Scenarios 1 and 2 for the period 1981–2099.

Figure 14 shows the average monthly unmet demand for short-, middle-, and long term. For both scenarios, the highest unmet demand occurs from May to October during the dry season, with peaks in July and August due to the lack of precipitation.

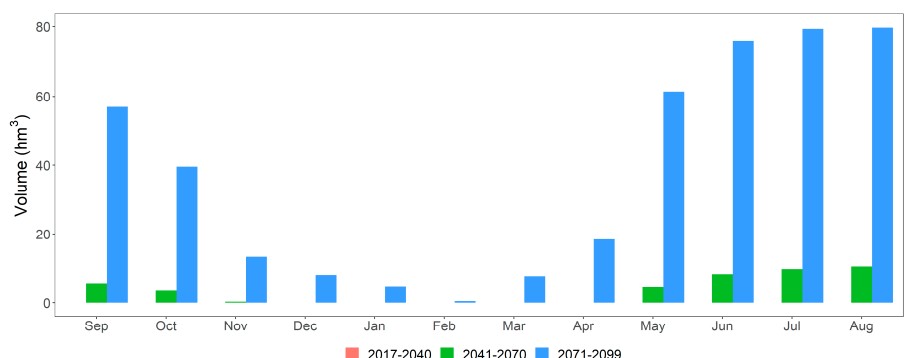

**Figure 14.** Monthly unmet demand for Scenario 1 in the VUB region for the period (2017–2099).

*3.5. Climate Change Scenario with Socio-Economic Changes*

The population and agricultural demand have the highest impact on water security; therefore, two new scenarios are proposed: Scenario 3 is based on Scenario 1 and Scenario 4 on Scenario 2, each with their changes in the estimation of future demand. To reduce population demand, a policy of planned migration to the city of Cusco (Sub-basin 1) could be applied, in addition to encouraging responsible consumption of water resources. This migration policy would reduce the growth rate to 0.3% [9] by the year 2050. In Sub-basin 4, the growth rate remained constant at 0.2%, and no further reduction is possible. By 2050, the irrigation infrastructure will have increased, reaching irrigation efficiencies of 0.8 in all cropping areas.

Figure 15 compares the four modeled scenarios for the short- (2017–2040), middle- (2041–2070), and long-term (2071–2099). The reduction in unmet demand can be seen in Scenario 3 and Scenario 4. For the middle term, the unmet demand in Scenario 3 is 34.6 hm$^3$, a reduction of 39%. Scenario 4 has an unmet demand of 26.7 hm$^3$, a reduction of 37%. For the long term, the unmet demand in Scenario 3 is 252.8 hm$^3$, a reduction of 46%. Scenario 4 has an unmet demand of 238.2 hm$^3$, a reduction of 47%.

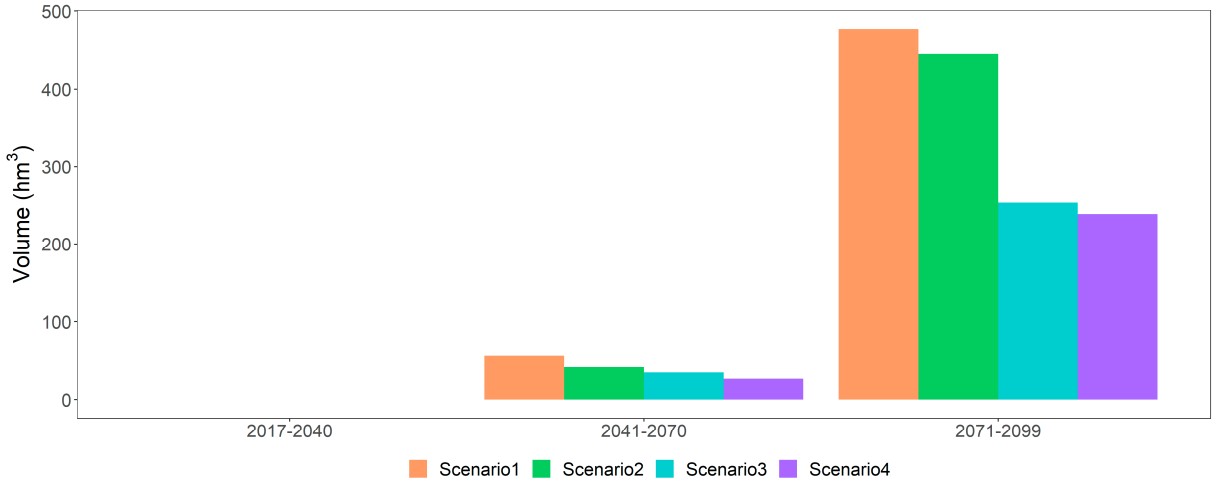

**Figure 15.** Annual unmet demand with socio-economic changes in Scenarios 1, 2, 3, and 4.

The actions to reduce the growth rate and increase irrigation efficiency mitigate the unmet demand but do not solve it completely. Although there is still a large water deficit, Sub-basin 4 has the highest unmet demand because it is in the headwaters of the VUB

and therefore does not receive the remainder. The unmet demand is lower in Sub-basin 1 because it receives all the contributions from the others.

## 4. Discussion

The Peruvian Andes lacks studies that include water planning and climate change projections to estimate water security. Only [31] which studied a Central Pacific basin and the present research which study a large Andean basin have used the WEAP model and climate change projections. In both studies, the WEAP model properly represents the hydrological processes. Furthermore, studies in the south American region [49,50] have demonstrated the value of the WEAP model in representing Andean catchment and water resource management. Hence, the WEAP model was chosen as the primary tool for carrying out the water balance.

Current demands were identified as population, agricultural, industrial, and energy demands. Population growth, current crop requirements, and industrial development were used for future projections. In total, 21 GCMs were used to simulate future water supply in contrast to 31 GCMs used by [31]. Variables held constant in the key assumptions could have a significant impact on water supply. However, to carry out projections of these variables, additional socioeconomic data is required as proxy variables. In a country with few available data such as Peru, it can yield great uncertainties in the projections.

There was a positive water balance for water use from 1987 to 2016. The amount of runoff and precipitation is 19 and 8 times higher than the demand. This would correspond to only 5.3% in terms of precipitation and 13% in terms of runoff. This surplus may allow demands to increase or intensify, e.g., by 10 to 15 times. However, availability is unevenly distributed in the Vilcanota-Urubamba basin. The highest availability is in the Vilcanota River in Sub-Basin 1, which is located at a lower altitude and receives runoff from the other sub-basins. For this reason, water allocation and water usage have been identified as the main factors driving water scarcity [10,14,51].

On an annual scale, climate change scenarios foresee an increase in rainfall, which leads to an increase in water availability, also observed by [14,31] for Andean catchments. Furthermore, the downscaled temperature projections preserve the warming trend [31,52,53]. However, the station distribution may not be well represented due to the lack of resolution of the model. Enhancing the resolution of the model is essential to improve the representation [9,14]. For the short term, the water supply is sufficient to cover the estimated demands for each sub-basin. From 2050 onwards, there will be an increase in unmet demand for each scenario. In the middle term, unmet demand is estimated at 56.72 hm$^3$ for Scenario 1 and 42.62 hm$^3$ for Scenario 2. For the long term, the unmet demand is 477.21 hm$^3$ in Scenario 1 and 445.80 hm$^3$ in Scenario 2. The study determined that reducing population growth (reducing rural–urban migration) and improving irrigation infrastructure (technification of irrigation systems) to increase efficiency would reduce unmet demand. For the middle term, the unmet demand in Scenario 3 is 34.6 hm$^3$, a reduction of 39%. Scenario 4 has an unmet demand of 26.7 hm$^3$, a decrease of 37%. For the long term, the unmet demand in Scenario 3 is 252.8 hm$^3$/year, a decrease of 46%. Scenario 4 has an unmet demand of 238.2 hm$^3$, a decrease of 47%.

The analysis through short-, middle-, and long term periods entails an uncertainty with each period. A main source of uncertainty is the multiple outputs after downscaling the processes of several climate models [41]. Although, we focused on short-, middle-, and long-term annual averages, extreme events, such as prolonged droughts, can have a profound effect on water availability. Another source of uncertainty is the limited capacity of GCMs to observe abrupt climate changes in the short- and middle term. Furthermore, the scenarios proposed in each period do not take into account social and economic changes that may occur in the long term.

According to the results, efforts in the region in implementing integrated water resource management should be substantially increased to avoid severe water scarcity. There are many challenges to water governance in a catchment relating to diverse interests and perceptions, as

well as power inequities among actors [10,14]. A water infrastructure investment will reduce water losses in the future, especially during drought conditions, and will counteract water demand growth. In line with this, natural infrastructure [54] and pre-Inca interventions [55], could be implemented to regulate water resources. These interventions would be ideal as they can be initiated and developed in the short term. Thus, during the middle- and long term, they would increase water availability in implemented locations.

## 5. Conclusions

This study used climate change scenarios to model the capacity of the VUB region to meet future water demand. The results showed that population growth would have a greater contribution than climate change impacts on future water stress. For the year 2050, an uncovered water demand is projected of between 56.72 hm$^3$ and 42.62 hm$^3$. The most pessimistic scenarios yield an unmet demand of 477.21 hm$^3$ by 2099. Furthermore, typical adaptation strategies such as migratory control and irrigation efficiency improvement were explored. It was shown that they are insufficient to offset the increased water demand. Thus, it is suggested that complementary adaptation strategies should be implemented to guarantee long-term water security.

Although using the best available data and reasonable scenario assumptions, different uncertainties contribute to the overall water balance estimates. The main uncertainties arise from climate change projections and the scarcity of hydrological observations. The model results would benefit from calibration and validation data at different spatial scales. Future studies could focus on better representation of the surface and groundwater interactions and the glacier contribution to streamflow.

Based on the results, we recommended the development of the capacity to monitor hydrological, climate, and water use. This development will be possible only with the involvement of government entities and the population. Population and agricultural growth will be the main increasing demands in the future; planning medium- and long term will be necessary to mitigate the impact on water security. To assist decision-makers, hydrological and social models must be developed using multiple scenarios. However, decisions at the local level must consider the variation in the spatiotemporal scale. Even though the study was conducted in the Andes of Peru, its findings may apply to other tropical Andean catchments.

**Author Contributions:** Conceptualization, A.G., P.R. and W.L.-C.; Funding acquisition, P.R.; Investigation, A.G.; Methodology, W.L.-C.; Project administration, P.R. and W.B.; Supervision, P.R., W.L.-C.; Validation, P.R.; Visualization, W.L.-C.; Writing—original draft, A.G.; Writing—review and editing, J.C.-A. and D.H. All authors have read and agreed to the published version of the manuscript.

**Funding:** This study was funded by the National Council for Science, Technology, and Technological Innovation (CONCYTEC) of Peru and the Newton Fund of England. N_ 005-2019-PROCIENCIA. Peru.

**Institutional Review Board Statement:** Not applicable.

**Informed Consent Statement:** Not applicable.

**Data Availability Statement:** All data, shapefile and model outputs will be made available on request to the correspondent author email with appropriate justification.

**Acknowledgments:** This research was conducted under the RAHU project "Water security and climate change adaptation in Peruvian glacier-fed river basins" or "seguRidad hídrica y Adaptación al cambio climático en cuencas Hídricas perUanas alimentadas por glaciares" (RAHU). The authors also thank CONCYTEC and Newton Fund for funding, as well as the RAHU project partners (Contract N_ 005-2019-PROCIENCIA. Peru) and PEGASUS project partners (Contract N_ 009-2019-PROCIENCIA. Peru) for providing information and expertise. We also thank the SEI (Stockholm Environmental Institute) for licensing the software to A.G.

**Conflicts of Interest:** The authors declare no conflict of interest.

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
