# Peer review of "Assessment of Present and Future Water Security under Anthropogenic and Climate Changes Using WEAP Model in the Vilcanota-Urubamba Catchment, Cusco, Perú"

_water, doi:10.3390/w15071439_

Round 1

Reviewer 1 Report

At the first I would like to thank Authors for hard work, which results is interesting manuscript. I have read article very carefully. The presented topic is very interesting and refers projected water management. The authors analyzed possibility of WEAP model using in the light of projected water demand for population, agriculture and industry. In general, article I prepared correctly. Nevertheless I had a some comments:

1.       Key words: please do not use abbreviation (WEAP).

2.       Lines 60-70: in my opinion this part is not necessary in Introduction.

3.       In introduction must be clearly highlighted what is the novelty of conducted work.

4.       Study area: please provide more information about meteorological factors and land use in basin.

5.       Data collection: please clearly define what kind of hydrological data were used in calculation.  

6.       Line 203: why the LULC changes were not considered? Land use is very important factor which shape water resources.

7.       Line 270: to the model calibration and validation the NSE and PBIAS was used. Please describe it in methodology.

8.       In discussion the Authors should provide some references.

9.       In the article is lack of conclusion.

Author Response

Response to Reviewer 1 Comments

At the first I would like to thank Authors for hard work, which results is interesting manuscript. I have read article very carefully. The presented topic is very interesting and refers projected water management. The authors analyzed possibility of WEAP model using in the light of projected water demand for population, agriculture and industry. In general, article I prepared correctly. Nevertheless I had a some comments:

Response 1:  We thank the Reviewer for his overall good opinion of our work and we hope this new version of the manuscript and our responses will satisfy his observations.

  1. Key words: please do not use abbreviation (WEAP).

Response 2: 

We have modified the text as follows:

“Water is an essential resource for social and economic development. The availability of this re-source is always threatened by the rapid increase in its demand. This research assesses current (2010-2016) and future (up to 2099) levels of water security considering socio-economic and climate change scenarios using the water evaluation and planning model (WEAP) model in Vilcano-ta-Urubamba (VUB) catchment.”

  1. Lines 60-70: in my opinion this part is not necessary in Introduction.

Response 3: It was corrected and reduced in the introduction section (see lines 59-69)

  1. In introduction must be clearly highlighted what is the novelty of conducted work.

Response 4:

We agree. We have modified the text as follows:

“Peru has a unique combination of rainfall variability and landscapes that generate different hydrologic conditions and water availability [6]. The capacity for cities to cope with climate change is further stressed by socioeconomic drivers such as population growth and water consumption intensity [7,8]. Although previous work has included climate change scenarios for future water security in the Andes [9–11], few of them have included socioeconomic drivers to assess if local adaptation strategies would be enough to guarantee future water demand [12]. A potential reason for the inexistence of this exercise is the data scarcity, both in terms of hydrometeorological and socioeconomical observations in the Andes. This has hindered the possibility to make accurate predictions about the future with complex hydrological models and has instead favored the use of simpler conceptual models [11].”

  1. Study area: please provide more information about meteorological factors and land use in basin.

Response 5:

We agree. We have modified the text as follows:
“Precipitation is highly seasonal, with the wet season from November to March and the dry season from June to August. During the wet season precipitation is between 120 to 150 mm and during the dry season it is 70 to 0 mm. Therefore, mean precipitation is 797 mm. Regarding the temperature, the minimum is from -5°C to 4°C, while the maximum is from 16°C to 19°C.

The most common ground cover is the high Andean grassland (56.9%), followed by bare soil (12.8%), scrub (11.4%) and agriculture (9.8%). The watershed has a broad upper part due to the intense erosion of glacial and alluvial origin. The lower part of the basin is narrower due to its lithology, which corresponds mainly to hard sedimentary rocks that do not allow lateral erosion processes, with vertical erosion predominating.”

Figure 2. The land cover/use map of the VUB catchment derived from the National Ecosystem Map data.

  1. Data collection: please clearly define what kind of hydrological data were used in calculation.

Response 6: 

We have modified the text as follows:

“Different data sources were used to develop the hydrological model (Table 2). The hydrologic data used in the calculation are precipitation, evapotranspiration, and stream-flow.”

  1. Line 203: why the LULC changes were not considered? Land use is very important factor which shape water resources.

Response 7: Thanks for the question, We are not considering land use and land cover changes because require additional socioeconomic data as proxy variables and a country with little available data such as Peru can yield considerable uncertainties in projections.

  1. Line 270: to the model calibration and validation the NSE and PBIAS was used. Please describe it in methodology.

Response 8:

This section was added

2.3.1 Model evaluation statistics

In the sensitivity analysis, two statistics were used to quantify how well the simulat-ed data

represent the observed data. The first is the Nash-Sutcliffe coefficient of efficiency (NSE)

Equation 1 and the second is the percentage deviation (PBIAS) Equation 2. Table 4 shows the

ranges of values used for the classification of the metrics used in each run ac-cording to [24].

Were Qoi is the ith flow observation, Qsi is the ith flow simulated, Qmi is mean of ob-served flow

data and n is the total number of observations.

Table 4. General performance ratings for statistics [24].

Performance rating

PBIAS (%)

NSE

Very good

PBIAS < ±10

0.75 < NSE ≤ 1.00

Good

±10 ≤ PBIAS < ±15

0.65 < NSE ≤ 0.75

Satisfactory

±15 ≤ PBIAS < ±25

0.50 < NSE ≤ 0.65

Unsatisfactory

PBIAS ≥ ±25

NSE ≤ 0.50

  1. In discussion the Authors should provide some references.

Response 9: 

We agree. We have modified the text as follows:

“The Peruvian Andean lack studies that use WEAP and climate change projections to estimate water security. Only [29] and this study have used WEAP and climate change projections to model a catchment, however, Olsson did so in a coastal river catchment in Lima. Nevertheless, in both studies, WEAP model represents properly the hydrological processes. Furthermore, studies in the south American region [46,47] have demonstrated the value of WEAP model in representing Andean catchment and water resources man-agement. Hence, WEAP model was the primary tool for carrying out the water balance.”

“Current demands were identified as population, agricultural, industrial, and energy demands. Population growth, current crop requirements, and industrial development were used for future projections. In total 21 GCMs were used to simulate future water supply in contrast of 31 GCMs used by [29].”

“Natural infrastructure [50] and pre-Inca interventions [51], in addition to managing socioeconomic factors, could be implemented to regulate water resources. These interventions would be ideal as they can be initiated and developed during the period of positive water balance (2017–2040). Thus, during the water deficit, they would reduce the unmet demand in the areas where they are implemented.”

  1. In the article is lack of conclusion.

Response 10: 

This section was added

  1. Conclusions

This study used climate change scenarios to model the capacity of the VUB watershed to meet future water demand. The results showed that population growth would have a greater contribution than climate change impacts on future water stress. The most pessimistic scenarios yield an unmet demand of 477.21 hm3 by 2099 while the rest scenarios are within an interval between 56.72 hm3 and Y hm3. Furthermore, typical adaptation strategies such as migratory control and irrigation efficiency improvement were explored. It was shown that they are insufficient to offset the increased water demand. This way, it is suggested that complementary adaptation strategies should be implemented to guarantee long term water security.

Although using best available data and reasonable scenario assumptions, different uncertainties contribute to the overall water balance estimates. Main uncertainties arise from climate change projections and the scarcity of hydrological observations. The model results would benefit from calibration and validation data at different spatial scales. Future studies could focus on better representing the surface and groundwater interactions and the glaciers contribution to streamflow.

Reviewer 2 Report

Dear authors,

I have gone through the manuscript entitled "Assessment of present and future water security under anthropogenic and climate changes using WEAP model in the Vilcanota-Urubamba catchment, Cusco, Perú", submitted to the Water (MDPI) Journal. Based on my review, I believe the paper has a strong potential for publication.

The content is sound, the methology is appropriate and supports well the results and findings presented. The figures and tables are very clean (for most of them) and the manuscript is pleasant to read and easily understable. Also, the topic is important regarding the context and in that regard, the paper provides very interesting outcomes for future planning.

However, sometimes, I found the paper to be a bit lenghty, especially in the Results section. Also, the Discussion focuses too much on the findings of this study, but does not adress the outside literature.

There are also minor comments I have laid out within the manuscript, to which I expect the authors to provide point by point answers.

I recommend Minor Revision. Congratulations for the interesting work.

Author Response

Response to Reviewer 2 Comments

Dear authors,

I have gone through the manuscript entitled "Assessment of present and future water security under anthropogenic and climate changes using WEAP model in the Vilcanota-Urubamba catchment, Cusco, Perú", submitted to the Water (MDPI) Journal. Based on my review, I believe the paper has a strong potential for publication.

The content is sound, the methology is appropriate and supports well the results and findings presented. The figures and tables are very clean (for most of them) and the manuscript is pleasant to read and easily understable. Also, the topic is important regarding the context and in that regard, the paper provides very interesting outcomes for future planning.

However, sometimes, I found the paper to be a bit lenghty, especially in the Results section. Also, the Discussion focuses too much on the findings of this study, but does not adress the outside literature.

There are also minor comments I have laid out within the manuscript, to which I expect the authors to provide point by point answers.

I recommend Minor Revision. Congratulations for the interesting work

Response 1:  We thank the Reviewer for his overall good opinion of our work and we hope this new version of the manuscript and our responses will satisfy his observations.

Below we present the modifications made for the indicated sections

1.Introduction: The following paragraph was added in the new version

Peru has a unique combination of rainfall variability and landscapes that generate different 

hydrologic conditions and water availability [6]. The capacity for cities to cope with climate change is further stressed by socioeconomic drivers such as population growth and water consumption intensity [7,8]. Although previous work has included climate change scenarios for future water security in the Andes [9–11], few of them have included socioeconomic drivers to assess if local adaptation strategies would be enough to guarantee future water demand [12]. A potential reason for the inexistence of this exercise is the data scarcity, both in terms of hydrometeorological and socioeconomical observa-tions in the Andes. This has hindered the possibility to make accurate predictions about the future with complex hydrological models and has instead favored the use of simpler conceptual models [11].

Commented [A10]: Please, add a footer to the table, where it can be specified that ee.Image functions are used within Google Earth Engine to access to specific data. 

Added in the new version

Commented [A11]: 1.Please write “Cropwat”, not “Cropwatt”. 2.The WEAP1 and WEAP2 mentions are not that clear and should be explained in the footer.

Added in the new version

Commented [A12]: Please provide some references for thesevalues. Do they come from national Plan/Strategy? Control policies? 

References for this values were added

Commented [A13]: Pleae provide additional details on the hydrology setup: soil types ? infiltration method ? Surface runoff method ?  Do not describe extensively these methods, but just rather say what were the methods used. 

The catchment simulation method is Rainfall Runoff method (Soil Moisture). This method is more complex, representing the catchment with two soil layers. For this method we characterize the land cover (Figure 2) with every parameter describe in Equation 1.

Commented [A14]: Great and honest! I do appreciate. In the discussion, please come back to these assumptions and provide incentives on how they are likely to affect the findings. 

Thanks for the good opinion, was added in the conclusions.

Commented [A15]: Something is not clear:

-Approach 1: Did you carry out individual future

simulation with every single model in the ensemble,

before averaging the outputs?

-Approach 2: Or did you evaluate first a “mean

model” and simulated the future water demand

using this mean model?

Well, if you did the first, that is definitely the best

option. If you did the second, well, that is a flawed

approach by all means, as a mean model cannot be

considered valid, since the discrepancy between all

models (in their initial form) is large, and conveys

actually an uncertainty which should be typically

reported. The only way to carry out that uncertainty to

the simulated outputs is use an ensemble of outputs, as

defined in approach 1. 

Thank you very much for your comment, the second approximation was used because it was used previously in an investigation in Peru. However, let us consider the first approximation for future research.

Commented [A16]: Not exactly the means, but raththe quantiles distributions are adjusted.

Please also consider that quantile mapping actually deteriorates long term trends. However, for the purpoof this study, it is simple and seems appropriate. 

Commented [A17]: Provide Percent Biases for calibration and validation periods here. 

Added in the new version

Commented [A18]: It might be very interesting to oppose these inflows/outflows to monthly rainfall (with vertical bars in reverse on a separate axis) 

Due to lack of time it could not be added to this publication, for future presentations it will be added.

Commented [A19]: Good. Provide p-values for panels a and b. 

Added in the new version

Commented [A20]: How was the shaded areas (in red and blue) calculated ? Is that the variability over all the model outputs? if so, make sure that it represents the 95% uncertainty band. 

Commented [A21]: Same question: How was the shaded areas (in red and blue) calculated ? Is that the variability over all the model outputs ? if so, make sure that it represents the 95% uncertainty band.

Shade areas represent the 95% uncertainty of the projected demand use in every scenario.

Commented [A22]: Also provide p-values. 

Added in the new version

Commented [A23]: Convoluted sentence.  It can simply be said that the increase irrigation efficiency mitigates the unmet demand, but does notsolve it completely. 

Corrected  in the new version

Commented [A24]: Not needed, just provide the number of climate models in the ensemble used. 

Corrected  in the new version

Commented [A25]: Again, avoid the use of “we” and use the passive form instead. 

Corrected  in the new version

4.Discussion

The Peruvian Andean lack studies that use WEAP and climate change projections to estimate water security. Only [29] and this study have used WEAP and climate change projections to model a catchment, however, Olsson did so in a coastal river catchment in Lima. Nevertheless, in both studies, WEAP model represents properly the hydrological processes. Furthermore, studies in the south American region [46,47] have demonstrated the value of WEAP model in representing Andean catchment and water resources man-agement. Hence, WEAP model was the primary tool for carrying out the water balance. 

Current demands were identified as population, agricultural, industrial, and energy demands. Population growth, current crop requirements, and industrial development were used for future projections. In total 21 GCMs were used to simulate future water sup-ply in contrast of 31 GCMs used by [29]. Variables held constant in the key assumptions could have a significant impact on water supply. However, to carry out projections of these variables, additional socioeconomic data is required as proxy variables. In a country with few available data such as Peru, it can yield to great uncertainties in the projections.

There was a positive water balance for water use from 1987 to 2016. The amount of runoff and precipitation is 19 and 8 times higher than the demands. This would corre-spond to only 5.3% in terms of precipitation and 13% in terms of runoff. This surplus may allow demands to increase or intensify, e.g., by 10 to 15 times. However, availability is unevenly distributed in the Vilcanota-Urubamba basin. The highest availability is in the Vilcanota River in Sub-Basin 1, which is located at a lower altitude and receives runoff from the other sub-basins.

Climate change scenarios predict an increase in precipitation and temperature. Fur-thermore, the downscaled temperature projections preserve the warming trend [29,48,49]. For the period 2017–2040, the water supply is sufficient to cover the estimated demands for each sub-basin. From 2050 on, there will be an increase in unmet demand for each scenario. In the period 2041–2070, unmet demand is estimated at 56.72 hm3 for Scenario 1 and 42.62 hm3 for Scenario 2. For the period 2071–2099, it is 477.21 hm3 in Scenario 1 and 445.80 hm3 in Scenario 2. The study determined that reducing population growth (reduc-ing rural-urban migration) and improving irrigation infrastructure (technified irrigation systems) to increase efficiency would reduce unmet demand. For the period 2041–2070, the unmet demand in Scenario 3 is 34.6 hm3, a reduction of 39%. Scenario 4 has an unmet demand of 26.7 hm3, a reduction of 37%. For the period 2070–2099, the unmet demand in Scenario 3 is 252.8 hm3/year, a reduction of 46%. Scenario 4 has an unmet demand of 238.2 hm3, a reduction of 47%.

Natural infrastructure [50] and pre-Inca interventions [51], in addition to managing socioeconomic factors, could be implemented to regulate water resources. These interventions would be ideal as they can be initiated and developed during the period of positive water balance (2017–2040). Thus, during the water deficit, they would reduce the unmet demand in the areas where they are implemented.

Reviewer 3 Report

General comments

The authors present a study entitled “Assessment of present and future water security under anthropogenic and climate changes using WEAP model in the Vilcanota-Urubamba catchment, Cusco, Peru.”. The work aims to “assesses current (2010-2016) and future (up to 2099) levels of water security considering socio-economic and climate change scenarios using the WEAP model in Vilcanota-Urubamba (VUB) catchment”. It is the opinion of this reviewer that the work, unfortunately, presents important deficiencies that negatively affect the achievement of the proposed objective satisfactorily. The main reasons are the following:

a) The Introduction section mainly presents a type of summary of the work, but marginally and weakly addresses the reasons that justify the study and the novelty of the research regarding the state of the art in water balance modeling issues. Questions such as, how important is the problem of water security in Peru, in general, and in the study area, in particular? What are the studies on the subject carried out to date, and with what metrics have they been evaluated? How has the lack of data influenced previous studies? Are studies that combine variables from very long-term climate models with projections of socio-economic variables in situ based on short records reliable? Is it possible to extrapolate the goodness-of-fit metrics of the balance model to sub-basins without gauging stations? In summary, the Introduction section should have explicitly raised pertinent problems in the study area that may be of interest in other regions due to the way in which the authors approached their solution. This key aspect is not present in this section.

b) The Materials and Methods section shows a significant number of inconsistencies and a lack of clarity and detail in the methodological aspects (see detailed comments). In concrete terms, the calibration/validation procedure is not clear, especially in terms of the recording periods used as well as the number and location of the calibration points. In this sense, a figure with the general scheme of the basin in WEAP was clearly missing, highlighting the points of interest for a better interpretation of the calibration/validation metrics. Another relevant aspect has to do with the available evidence to support the creation of scenarios of almost 80 years, based on no more than 10 or 20 years of records for key water balance variables. This is a case of extrapolation with a clear level of uncertainty that is not properly included in the study and is relevant to the interpretation of the results. There are, in this regard, many assumptions with very little supporting evidence.

c) The results section is confusing and shows inconsistencies between the explanation and supporting figures or tables that negatively affect an adequate interpretation of the results. There is an absence of metrics that are key to a better interpretation of the dynamics of water security throughout the projected period. Unfortunately, the use of long-term cumulative measures prevents appreciating the effect of short-term events (such as droughts of various durations) that are much more relevant to water security than aggregate statistics such as those presented in the study. For example, what percentage of the projected time is it possible to find events where the coverage of a demand node was less than a certain threshold value? How often do water deficit events occur that last 3, 6, or even 12 consecutive months? .

d) Finally, the Discussion section barely mentions what was done without making relevant comparisons with similar studies in the study area or similar regions, it does not address the key questions that should have been raised in the Introduction section.

In summary, the manuscript shows significant deficiencies for this type of work, despite the relevance that the analysis of water security implies in contexts of scarcity of data. It is hoped that these comments will allow authors to improve future versions of their work.

Detailed comments

Abstract:

Lines 26-27: Please consider change the sentence to “were used for calibrate (1987-2006) and validate (2007-2016) a Weap Model for the VUB catchment.”

Introduction:

Line 49: Please consider to complement the “ecological stability” term with “environmental sustainability” as the former show a more restricted meaning than the second one.

Line 53: Please fix “two significant signs of climate change a global” to “two significant signs of climate change:  global”

Line 60: Please add “southeastern Peru” after “Cusco región.”

Line 64: Please make appropiated use of superscripts in km2

Line 65: Please make appropiated use of superscripts in m3/s

Line 77: Please add more references and include more detail to support the maim idea of risk in wáter supply for the population as this assumption behind the study and its justification. Which is the population growth rate in the área?, Is greater than other places in the country?, why the concern about difficulties to meet water demand requeriments in this region?

Line 76: Plase change “will be used for existing”  to “will be used for modeling the existing”.

General comment: Please consider to include major modifications to this section. The main idea of the introducton section is to present the main problema, hypothesis and justification of the current study, not just present and extended symmary of the work focused on what was done. Many questions regard to the justification of the research where not addressed properly: for example, what is the importance of the study area in the context of water scarcity and water security problems in Peru?, What ar the main limitations in current research in the VUB catchment that proposed study wants to address?. Why is necessary to project the timeline of the study up to 2099 if no equivalent projection is done in terms of water demand with the same accuracy or reliability in model’s performace?.  In resume, please improve the Introduction section and study justification proving that the choosen methodology provides appropiate answers to relevant and pertinent questions.

Material and methods

Study area

Line 104: Please fix to “The study area is localed in …”

Line 108: Please make appropiated use of superscripts in km2

Lines 116-117: Please modify Figure 1. Many places mentioned in the text are not shown properly in the map. Similarily, the map show some terms (Pitumarca, Sicuani, Sibinacocha) not detailed in the text or map’s legend.

Line 129: Please fix “the flow data” to “the streamflow data”

Line 131: Please fix “Daily observed flows“ to “Daily streamflow record was collected”.

Line 134: Please fix “Population numbers” to “ population statistics”. At the same time, please indicate the date of the reference “INEI’s Peruvian National Census”.

Methodology

Line 153: Please explain and fix if needed. Authors indicate that “Calibration was carried out from 1987 to 2006, and validations from 153 2007 to 2016.” . However, according to Lines 132-133, streamflow data was available only for period 2010-2016. What data was used for calibration and validation?

Lines 152-158: The authors provide few justification and support for the choosen values used to create their scenarios. Is reliable the assumption of a fixed population growth of 1.8% annually up to 2099?. What kind of evidence can support this assumption? The same for irrigation efficiency. Please improve the justification of the choosen values and scenario building.

Hydrological Model

Lines 175-184:  I recommend reduce this section which provides excess of details of equations of a well-known and broadly used model and, instead that, provide more detail on which datasets and techniques were used to feed the Weap Model.

 Line 184-185: The quality of Figure 3 must be improved.

Line 191: Contrary to what was stated before, here the authors indicate that the calibration period is from 2010-2016. However, according to Lines 131-132, this is the same period of observed records, so no data for validation is considered. Please fix, improve and correct all necessary sections to clarify how the model was calibrated and validated.

Lines 194-197: Please provide adequate evidence, references and/or criterio used to support or justify these assumptions.

Line 228: Please provide detail about how GCM data was used to estimate future water demand. In particular, which GCM data?, And how GCM data was transformed in water demand?

Line 230: Please fix equation number

Line 235: Please indicate what is “RADA” acronym.

Lines 253-260: Please add a new subsection 2.8 and better explain the statistical downscaling procedure.

Results

3.1. Model calibration and validation

Lines 263-264: Again, there are inconsistencies  in the the description of calibration and validation periods

Lines 264-267: Please refer only to metrics given in Table 5. Do not include comments (like “underestimate (30%)) out of Table 5.

Line 267: Please fix vertical axis label in Figure 5. The plot shows monthly streamflow time series, not runoff or flow.

Lines 276-279: As Table 5 only refers to GOF metrics for one point (outlet), there is no confidence about the performance of the model in other points of the catchment. How can the authors support the main idea of “high degree of confidence of the prediction”?.

3.2 Initial demand and water secutiry

Line 282: Please fix Table number. Must be Table 6 instead Table 5.

Lines 283-286: Please make appropiated use of superscripts and units in hm3/year

Lines 288-289: No data about simulated (Qs) or observed(Qo) is presented in Table 6.

 Line 285: Please make appropiated use of superscripts and units in hm3/year

Line 291: Please make appropiated use of superscripts and units in hm3/year

Line 294: Please make appropiated use of superscripts and units in hm3

Line 295: Please make appropiated use of superscripts and units in hm3

Lines 302-304: No spatially distributed data is presented to support the idea of “equally distributed in the VUB”. Authors must consider to include a scheme or map with relevant model outpus.

3.3.1 Population demand

Line 314: Authors mention equation 1 for growth rate estimation, but clearly equation 2 is the correct one. In addition,    no detail about the source of data for the water demand versus population relationship presented in Figure 7 was provided. At the same time, there is no explanation to understand why plots only show relationships for sub-basins 1 and 4 and not other sub-basins are included.

Lines 319-321: The population demand curves are unrealistics and estimated values are unreliable based on the used data sources and associated methodology.

3.3.2. Agricultural demand

Lines 335-343: This section must be improved entirely. Authors must avoid the use od acronym (like P and T) and refers to precipitation and temperature. Additionally, scenario specifications must also be detailed again for proper interpretation of the results.

Lines 340-342: Please make appropiated use of superscripts and units in hm3

3.3.3. Industrial demand

Line 350: Please make appropiated use of superscripts and units in hm3

Line 350: Authors mention projection for industrial demand up to 2099 but Figure 10 only show date values up to 2016. Please correct this kind of inconsistencies.

 3.3.4 Future scenario in the context of climate change (CC)

Lines 355-356: Please make appropiated use of superscripts and units in hm3

Line 358: Authors make statements about sub-basins 1 and 4 in reference to Figure 11 but the plot have no data with regard to the above mentioned sub-basins. Please fix.

Line 365: Authors make statements about “both scenarios” but Figure 12 only show data for Scenario 1. Please clarify.

3.3.4. C scenario with socio-economic changes

Line 376: Please explain why if evidence or available data can support population growth rate up to 2050, scenarios extrapolate to 2099. What is the criterio used tu support this assumption?

Lines 383-384: Please make appropiated use of superscripts and units in hm3

Line 393: Figure caption show clearly mistakes in period specification (2050-1999). Please fix.

Lines 399-403: Please improve this section. What is the value of listing all GCMs?

4. Discussion

Lines 410-416: Please make appropiated use of superscripts and units in hm3

Author Response

Response to Reviewer 3 Comments

The authors present a study entitled “Assessment of present and future water security under anthropogenic and climate changes using WEAP model in the Vilcanota-Urubamba catchment, Cusco, Peru.”. The work aims to “assesses current (2010-2016) and future (up to 2099) levels of water security considering socio-economic and climate change scenarios using the WEAP model in Vilcanota-Urubamba (VUB) catchment”. It is the opinion of this reviewer that the work, unfortunately, presents important deficiencies that negatively affect the achievement of the proposed objective satisfactorily. The main reasons are the following:

  1. a) The Introduction section mainly presents a type of summary of the work, but marginally and weakly addresses the reasons that justify the study and the novelty of the research regarding the state of the art in water balance modeling issues. Questions such as, how important is the problem of water security in Peru, in general, and in the study area, in particular? What are the studies on the subject carried out to date, and with what metrics have they been evaluated? How has the lack of data influenced previous studies? Are studies that combine variables from very long-term climate models with projections of socio-economic variables in situ based on short records reliable? Is it possible to extrapolate the goodness-of-fit metrics of the balance model to sub-basins without gauging stations? In summary, the Introduction section should have explicitly raised pertinent problems in the study area that may be of interest in other regions due to the way in which the authors approached their solution. This key aspect is not present in this section.

Response 1:  We thank the Reviewer for his overall good opinion of our work and we hope this new version of the manuscript and our responses will satisfy his observations. The justification and novelty of the research were added. Peru suffers from water scarcity in semi-arid regions where population is concentrated as the case of the Vilcanota-Urubamba basin (VUB) with its main city Cusco, a widely visited touristic place. The novelty is the study for the first time in VUB about the impact of climate change in a mountainous water security context and data scarcity. The selection of diverse methods and tools used in this research will help the reader as a guideline for this type of studies.

  1. b) The Materials and Methods section shows a significant number of inconsistencies and a lack of clarity and detail in the methodological aspects (see detailed comments). In concrete terms, the calibration/validation procedure is not clear, especially in terms of the recording periods used as well as the number and location of the calibration points. In this sense, a figure with the general scheme of the basin in WEAP was clearly missing, highlighting the points of interest for a better interpretation of the calibration/validation metrics. Another relevant aspect has to do with the available evidence to support the creation of scenarios of almost 80 years, based on no more than 10 or 20 years of records for key water balance variables. This is a case of extrapolation with a clear level of uncertainty that is not properly included in the study and is relevant to the interpretation of the results. There are, in this regard, many assumptions with very little supporting evidence.

Response 2: All inconsistencies were fixed in the new version. Figure 5 shows the weap scheme of the basin. Where the points of water demand in each sub-basin are shown. Due to the lack of estimates for the future, the most relevant variables should be assumed. These assumptions allow us to estimate at least one probable scenario for the future.

  1. c) The results section is confusing and shows inconsistencies between the explanation and supporting figures or tables that negatively affect an adequate interpretation of the results. There is an absence of metrics that are key to a better interpretation of the dynamics of water security throughout the projected period. Unfortunately, the use of long-term cumulative measures prevents appreciating the effect of short-term events (such as droughts of various durations) that are much more relevant to water security than aggregate statistics such as those presented in the study. For example, what percentage of the projected time is it possible to find events where the coverage of a demand node was less than a certain threshold value? How often do water deficit events occur that last 3, 6, or even 12 consecutive months?.

Response 3:  Water security was based on a hydrological balance approach. It was not intended to evaluate occurrence or duration of deficit and excess.

  1. d) Finally, the Discussion section barely mentions what was done without making relevant comparisons with similar studies in the study area or similar regions, it does not address the key questions that should have been raised in the Introduction section.

Response 4:  We agree. We have modified the text as follows:

“The Peruvian Andean lack studies that use WEAP and climate change projections to estimate water security. Only [29] and this study have used WEAP and climate change projections to model a catchment, however, Olsson did so in a coastal river catchment in Lima. Nevertheless, in both studies, WEAP model represents properly the hydrological processes. Furthermore, studies in the south American region [46,47] have demonstrated the value of WEAP model in representing Andean catchment and water resources man-agement. Hence, WEAP model was the primary tool for carrying out the water balance.”

“Climate change scenarios predict an increase in precipitation and temperature. Furthermore, the downscaled temperature projections preserve the warming trend [29,48,49].“

“Natural infrastructure [50] and pre-Inca interventions [51], in addition to managing socioeconomic factors, could be implemented to regulate water resources.”

In summary, the manuscript shows significant deficiencies for this type of work, despite the relevance that the analysis of water security implies in contexts of scarcity of data. It is hoped that these comments will allow authors to improve future versions of their work.

 Detailed comments

Response 4: All the comments and suggestions were included as much as possible in the new version. The authors would like to thank you for your effort in conducting this review, which will be of great assistance in future research.

 Abstract:

Lines 26-27: Please consider change the sentence to “were used for calibrate (1987-2006) and validate (2007-2016) a Weap Model for the VUB catchment.”

Changed in the abstract

 Introduction:

Line 49: Please consider to complement the “ecological stability” term with “environmental sustainability” as the former show a more restricted meaning than the second one.

We agree. We replace it.

Line 53: Please fix “two significant signs of climate change a global” to “two significant signs of climate change:  global”

We fixed thy typo.

 Line 60: Please add “southeastern Peru” after “Cusco región.”

 Line 64: Please make appropiated use of superscripts in km2

We fixed thy typo.

Line 65: Please make appropiated use of superscripts in m3/s

We fixed thy typo.

Line 77: Please add more references and include more detail to support the maim idea of risk in wáter supply for the population as this assumption behind the study and its justification. Which is the population growth rate in the área?, Is greater than other places in the country?, why the concern about difficulties to meet water demand requeriments in this region?

Added in the new introduction

Line 76: Plase change “will be used for existing”  to “will be used for modeling the existing”.

We fixed thy typo.

General comment: Please consider to include major modifications to this section. The main idea of the introducton section is to present the main problema, hypothesis and justification of the current study, not just present and extended symmary of the work focused on what was done. Many questions regard to the justification of the research where not addressed properly: for example, what is the importance of the study area in the context of water scarcity and water security problems in Peru?, What ar the main limitations in current research in the VUB catchment that proposed study wants to address?. Why is necessary to project the timeline of the study up to 2099 if no equivalent projection is done in terms of water demand with the same accuracy or reliability in model’s performace?.  In resume, please improve the Introduction section and study justification proving that the choosen methodology provides appropiate answers to relevant and pertinent questions.

 Material and methods

Study area

Line 104: Please fix to “The study area is localed in …”

We fixed thy typo.

Line 108: Please make appropiated use of superscripts in km2

We fixed thy typo.

Lines 116-117: Please modify Figure 1. Many places mentioned in the text are not shown properly in the map. Similarily, the map show some terms (Pitumarca, Sicuani, Sibinacocha) not detailed in the text or map’s legend.

Line 129: Please fix “the flow data” to “the streamflow data”

We fixed the typo.

Line 131: Please fix “Daily observed flows“ to “Daily streamflow record was collected”.

We fixed the typo.

 Line 134: Please fix “Population numbers” to “ population statistics”. At the same time, please indicate the date of the reference “INEI’s Peruvian National Census”.

We fixed the typo.

 Methodology

Line 153: Please explain and fix if needed. Authors indicate that “Calibration was carried out from 1987 to 2006, and validations from 153 2007 to 2016.” . However, according to Lines 132-133, streamflow data was available only for period 2010-2016. What data was used for calibration and validation?

We fixed the typo. streamflow data was available only for period 1987-2016

Lines 152-158: The authors provide few justification and support for the choosen values used to create their scenarios. Is reliable the assumption of a fixed population growth of 1.8% annually up to 2099?. What kind of evidence can support this assumption? The same for irrigation efficiency. Please improve the justification of the choosen values and scenario building.

 References were added to support the assumptions.

 Hydrological Model

Lines 175-184:  I recommend reduce this section which provides excess of details of equations of a well-known and broadly used model and, instead that, provide more detail on which datasets and techniques were used to feed the Weap Model.

 Line 184-185: The quality of Figure 3 must be improved.

Figure quality was improve

Line 191: Contrary to what was stated before, here the authors indicate that the calibration period is from 2010-2016. However, according to Lines 131-132, this is the same period of observed records, so no data for validation is considered. Please fix, improve and correct all necessary sections to clarify how the model was calibrated and validated.

We fixed thy typo.

Lines 194-197: Please provide adequate evidence, references and/or criterio used to support or justify these assumptions.

This study focuses on population growth and climate change because these, alone with land use and land cover change, are the biggest pressures in the hydrological cycle in the Andes [17]. We due to the lack of socioeconomic data available, land use and land cover changes are note considered. This can lead to considerable uncertainty in projections, especially in countries such as Perú that have limited data available.

Line 228: Please provide detail about how GCM data was used to estimate future water demand. In particular, which GCM data?, And how GCM data was transformed in water demand?

Data from water use licenses were used to estimate future water demand. These licenses come from a government entity that regulates the use of all water sources. No GCM data was used to estimate water demand

Line 230: Please fix equation number

We fixed the typo.

Line 235: Please indicate what is “RADA” acronym.

We have modified the text as follows: “Administrative Registry of Water Use Rights (RADA in Spanish)”

Lines 253-260: Please add a new subsection 2.8 and better explain the statistical downscaling procedure.

We added the new subsection.

 Results

3.1. Model calibration and validation

Lines 263-264: Again, there are inconsistencies  in the the description of calibration and validation periods

We fixed the typo.

 Lines 264-267: Please refer only to metrics given in Table 5. Do not include comments (like “underestimate (30%)) out of Table 5.

 Line 267: Please fix vertical axis label in Figure 5. The plot shows monthly streamflow time series, not runoff or flow.

We fixed the typo.

Lines 276-279: As Table 5 only refers to GOF metrics for one point (outlet), there is no confidence about the performance of the model in other points of the catchment. How can the authors support the main idea of “high degree of confidence of the prediction”?.

Due to the lack of other hydrometric stations, only this exit point could be used. However, we think that it does have a high level of confidence due to the validation performed on the model and the small area of the basin.

 3.2 Initial demand and water secutiry

Line 282: Please fix Table number. Must be Table 6 instead Table 5.

We fixed the typo.

Lines 283-286: Please make appropiated use of superscripts and units in hm3/year

We fixed the typo.

Lines 288-289: No data about simulated (Qs) or observed(Qo) is presented in Table 6.

We fixed the typo.

 Line 285: Please make appropiated use of superscripts and units in hm3/year

We fixed the typo.

Line 291: Please make appropiated use of superscripts and units in hm3/year

We fixed the typo.

Line 294: Please make appropiated use of superscripts and units in hm3

We fixed the typo.

Line 295: Please make appropiated use of superscripts and units in hm3

We fixed the typo.

Lines 302-304: No spatially distributed data is presented to support the idea of “equally distributed in the VUB”. Authors must consider to include a scheme or map with relevant model outpus.

Describing the distribution of water resources in the basin is not part of the objectives of the study.

 3.3.1 Population demand

Line 314: Authors mention equation 1 for growth rate estimation, but clearly equation 2 is the correct one. In addition,    no detail about the source of data for the water demand versus population relationship presented in Figure 7 was provided. At the same time, there is no explanation to understand why plots only show relationships for sub-basins 1 and 4 and not other sub-basins are included.

We fixed the typo for the equation. We don't include other sub-basins because approximately 99% of the entire population of the basin is concentrated in sub-basins 1 and 4, therefore, only these were taken into account for the analysis. 

Lines 319-321: The population demand curves are unrealistics and estimated values are unreliable based on the used data sources and associated methodology.

The aim of the study is to use official government data. So that the results represent the reality of the place, however, we will take your appreciation into account for future research.

3.3.2. Agricultural demand

Lines 335-343: This section must be improved entirely. Authors must avoid the use od acronym (like P and T) and refers to precipitation and temperature. Additionally, scenario specifications must also be detailed again for proper interpretation of the results.

We fixed the typo. Additionally, details for every scenario were added.

Lines 340-342: Please make appropiated use of superscripts and units in hm3

We fixed the typo.

3.3.3. Industrial demand

Line 350: Please make appropiated use of superscripts and units in hm3

We fixed the typo.

Line 350: Authors mention projection for industrial demand up to 2099 but Figure 10 only show date values up to 2016. Please correct this kind of inconsistencies.

New Figure 13 was added

3.3.4 Future scenario in the context of climate change (CC)

Lines 355-356: Please make appropiated use of superscripts and units in hm3

We fixed the typo.

Line 358: Authors make statements about sub-basins 1 and 4 in reference to Figure 11 but the plot have no data with regard to the above mentioned sub-basins. Please fix.

We fixed the typo.

Line 365: Authors make statements about “both scenarios” but Figure 12 only show data for Scenario 1. Please clarify.

We fixed the typo.

3.3.4. C scenario with socio-economic changes

Line 376: Please explain why if evidence or available data can support population growth rate up to 2050, scenarios extrapolate to 2099. What is the criterio used tu support this assumption?

Due to the fact that there is no previous research in relation to population growth projections, the current growth was maintained as a possible scenario in the future.

Lines 383-384: Please make appropiated use of superscripts and units in hm3

We fixed the typo.

Line 393: Figure caption show clearly mistakes in period specification (2050-1999). Please fix.

We fixed the typo.

Lines 399-403: Please improve this section. What is the value of listing all GCMs?

We fixed the typo.

  1. Discussion

Lines 410-416: Please make appropiated use of superscripts and units in hm3

We fixed the typo.

Reviewer 4 Report

The paper titled” Assessment of present and future water security under anthropogenic and climate changes using WEAP model in the Vilcanota-Urubamba catchment, Cusco, Perú” provides research about groundwater resources protection. The authors compare meteorological and hydrometric data to investigate the future water conditions. RCP 4.5 and 8.5 scenarios were applied under the presence and absence of management conditions. The authors applied WEAR (Water Evaluation and Planning) model to represent the water supply of the basin.

My recommendation is major revisions.

General comments

-Avoid using keywords that are already in the title.

-Study area: a lot of data are missing. Add information about the aquifers and hydrogeological conditions in the area.

-Why the WEAP model was chosen?

-The methodology is clearly analyzed.

-Figure4. Confused the reader. Please simplify it/ or divided into 2 figures.

-Discussion needs improvements. Please rewrite it from zero. Check the suggested literature below for help.

-Conclusions are missing.

Suggested literature

Modeling groundwater and surface water interaction: An overview of current status and future challenges (2022). Science of The Total Environment. 846, 157355.

Simulating future groundwater recharge in coastal and inland catchments. Water Resource Management.

Author Response

Response to Reviewer 4 Comments

The paper titled” Assessment of present and future water security under anthropogenic and climate changes using WEAP model in the Vilcanota-Urubamba catchment, Cusco, Perú” provides research about groundwater resources protection. The authors compare meteorological and hydrometric data to investigate the future water conditions. RCP 4.5 and 8.5 scenarios were applied under the presence and absence of management conditions. The authors applied WEAR (Water Evaluation and Planning) model to represent the water supply of the basin.

My recommendation is major revisions.

Response 1:  We thank the Reviewer for his overall good opinion of our work and we hope this new version of the manuscript and our responses will satisfy his observations.

General comments

-Avoid using keywords that are already in the title.

Response 2:  We agree. We have modified the text as follows:

“WEAP model, water balance, water security, climate change, Andean basin, Hydrological modelling”

-Study area: a lot of data are missing. Add information about the aquifers and hydrogeological conditions in the area.

Response 3: We agree. We add this description:

“The study area presents diverse hydrogeology. But the most representative is

fractured aquifers. These aquifers are characterized by quartzite, sandstone, shale, and silt clay.

There is a flow between fractured and deep flows. The type of aquifer is the fissured sedimentary

type [17].”

-Why the WEAP model was chosen?

Response 4: Add in new version

WEAP ("Water Evaluation and Planning") is a water allocation model that provides a

comprehensive approach to water resource planning [25]. The model relies on the water balance to

replicate the behavior of the hydrological cycle [26]. It estimates water resources by integrating

hydrology, land use, hydrogeology, climate, water quality, and anthropogenic effects [27]. Lines

168-173)

Furthermore, studies in the south American region [47,48] have demonstrated the value of WEAP

model in representing Andean catchment and water resources management. Hence, WEAP model

was the primary tool for carrying out the water balance.  (Lines 418-420)

-The methodology is clearly analyzed.

-Figure4. Confused the reader. Please simplify it/ or divided into 2 figures.

Response 5: Corrected

-Discussion needs improvements. Please rewrite it from zero. Check the suggested literature below for help.

Response 6: We agree. We have modified the text.

-Conclusions are missing.

Response 10: 

This section was added

  1. Conclusions

This study used climate change scenarios to model the capacity of the VUB watershed to meet future water demand. The results showed that population growth would have a greater contribution than climate change impacts on future water stress. The most pessimistic scenarios yield an unmet demand of 477.21 hm3 by 2099 while the rest scenarios are within an interval between 56.72 hm3 and Y hm3. Furthermore, typical adaptation strategies such as migratory control and irrigation efficiency improvement were explored. It was shown that they are insufficient to offset the increased water demand. This way, it is suggested that complementary adaptation strategies should be implemented to guarantee long term water security.

Although using best available data and reasonable scenario assumptions, different uncertainties contribute to the overall water balance estimates. Main uncertainties arise from climate change projections and the scarcity of hydrological observations. The model results would benefit from calibration and validation data at different spatial scales. Future studies could focus on better representing the surface and groundwater interactions and the glaciers contribution to streamflow.

Round 2

Reviewer 1 Report

I would like to thank Authors for manuscript improving. All my concerns were solved. In my opinion the manuscript can be accept in present form.

Author Response

We would like to thank the reviewer for accepting the article and for all the valuable comments and suggestions provided.

Reviewer 3 Report

General comments

The authors present a second version of the study entitled “Assessment of present and future water security under anthropogenic and climate changes using WEAP model in the Vilcanota-Urubamba catchment, Cusco, Peru.”

This second version shows significant improvements compared to the first. However, there are still some relevant aspects that should be addressed or incorporated in greater depth and detail in the manuscript to consider it suitable for publication.

In particular, the work offers an analysis of scenarios in the water management of the study basin considering climate factors and wáter use efficiency. In this sense, one of the main weaknesses of the work that is still maintained is that the novelty of the study is not clear or explicit. It does not offer new metrics for estimating water security, the use of WEAP and water management scenarios is widely known, as is the choosen downscaling procedure. On the other hand, the work assumes a long-term period in the analysis of climate change (up to 2099) without considering or mentioning the existence of other modes of climate variability on a decadal or multidecadal scale that may be even more relevant for water security. Last and not least, the discussion section does not compare the findings of the work with those obtained in other studies, both in the study area and in Peru and other similar regions, which serves at least to highlight the novelty of the research.

Based on the above mentioned comments, it is required that the introduction and discussion sections be significantly improved considering the aspects indicated.

Detailed comments

Abstract: Please fix units of hm3 in Abstract.

Line 299: Use lowercase r instead R, as growth rate symbol.

Line 275: Please delete the dot in “ transform [41]. and if the distribution“

Line 312: Please fix hm3

Line 347: Please use the same term in both units (l/person/day  versus l/inhabit/day

Line 372: PLease fix the axis of the plot to hm3 instead m3. And improve the resolution of the plot. The quality of the text it is not Good enough.

Line 394: Please use Climate Change instead CC in subsetion 3.3.4.

4. Discussion section: Please include as part of the discusión section, some aspects that are relevant in this kind of study but were not included, and are necessary to be aknowledge by policy makers and wáter managers. For example, the presence of abrupt climate change in short and middle terms instaed long term, decadal and multidecadal climate variablity not properly observed in GCM simulations, and other social and economic factors that can be change in the long term not considered in the proposed scenarios.

Author Response

The authors present a second version of the study entitled “Assessment of present and future water security under anthropogenic and climate changes using WEAP model in the Vilcanota-Urubamba catchment, Cusco, Peru.”

This second version shows significant improvements compared to the first. However, there are still some relevant aspects that should be addressed or incorporated in greater depth and detail in the manuscript to consider it suitable for publication.

In particular, the work offers an analysis of scenarios in the water management of the study basin considering climate factors and wáter use efficiency. In this sense, one of the main weaknesses of the work that is still maintained is that the novelty of the study is not clear or explicit. It does not offer new metrics for estimating water security, the use of WEAP and water management scenarios is widely known, as is the choosen downscaling procedure. On the other hand, the work assumes a long-term period in the analysis of climate change (up to 2099) without considering or mentioning the existence of other modes of climate variability on a decadal or multidecadal scale that may be even more relevant for water security. Last and not least, the discussion section does not compare the findings of the work with those obtained in other studies, both in the study area and in Peru and other similar regions, which serves at least to highlight the novelty of the research.

Based on the above mentioned comments, it is required that the introduction and discussion sections be significantly improved considering the aspects indicated.

Response 1:  

We appreciate the comments and contributions given. For this new version, the introduction and the discussion section have been improved as suggested. The justification and novelty of the research were added in line 74-76. It has been specified that the focus of the future study has been carried out in 3 periods: short, medium and long term.

We hope this new version of the manuscript and our responses will satisfy his observations of him.

Detailed comments

Abstract: Please fix units of hm3 in Abstract.

We fixed thy typo.

Line 299: Use lowercase r instead R, as growth rate symbol.

We fixed thy typo.

Line 275: Please delete the dot in “ transform [41]. and if the distribution“

We fixed thy typo.

Line 312: Please fix hm3

We fixed thy typo.

 Line 347: Please use the same term in both units (l/person/day  versus l/inhabit/day

We fixed thy typo.

Line 372: PLease fix the axis of the plot to hm3 instead m3. And improve the resolution of the plot. The quality of the text it is not Good enough.

We agree. We replace it.

Line 394: Please use Climate Change instead CC in subsetion 3.3.4.

We fixed thy typo.

  1. Discussion section: Please include as part of the discusión section, some aspects that are relevant in this kind of study but were not included, and are necessary to be aknowledge by policy makers and wáter managers. For example, the presence of abrupt climate change in short and middle terms instaed long term, decadal and multidecadal climate variablity not properly observed in GCM simulations, and other social and economic factors that can be change in the long term not considered in the proposed scenarios.

Response 2:  

Added in the discussion section:

The analysis through short, middle, and long terms periods entails an uncertainty with each period. A main source of uncertainty is the downscaling method, which does not account for potential changes in extreme events [7]. Although we focused on short, middle, and long term annual averages, extreme events, such as prolonged droughts, can have a profound effect on water availability. Another source of uncertainty, is the limited capacity of GCMs to observe abrupt climate changes in the short and middle term. Furthermore, the scenarios proposed in each period do not take into account social and economic changes that may occur in the long term.

Reviewer 4 Report

I have read the revised manuscript which is ready for publication

Author Response

(The authors gave the same response as above.)

Round 3

Reviewer 3 Report

Assessment of present and future water security under anthro- pogenic and climate changes using WEAP model in the Vil- canota-Urubamba catchment, Cusco, Perú.

General comments

The authors present a fourth version of the work entitled "Assessment of present and future water security under anthropogenic and climate changes using WEAP model in the Vilcanota-Urubamba catchment, Cusco, Peru.".

The current version has significantly improved the quality of the work, allowing a better understanding of its purpose, scope and interpretation of results by the authors. There are still some minor details to be corrected, but the current version adequately meets the requirements for publication in MDPI Water.

Detailed comments

Abstract

Lines 24-25: Please change “ water evaluation and planning model (WEAP) model in Vilcanota-Urubamba” to “Water Evaluation and Planning System (WEAP) in Vilcanota-Urubamba” as the last is the correct software name (https://www.weap21.org/WebHelp/index.html)

Line 27: Please change “The Nash Sutcliffe values for the calibration is 0.60. The Nash Sutcliffe 27 values for the validation period is 0.84.” to  “The Nash Sutcliffe efficiency was 0.60  and 0.84 for calibration and validation, respectively.”

 Line 40: Please change “This socio-economics” to “ This socio-economic”.

 Introduction

 Line 62: Please change  “impact on socioeconomic.” in “impact on socioeconomic activities”.

 Line 70: Please change “future water demand [14].” To “future water demand satisfaction [14].”

 Line 71: Please change “is the data scarcity” to “is data scarcity”.

 Discussion

Line 452: Please consider to change “From 2050 on”  to “From 2050 onwards”

Author Response

(The authors gave the same response as above.)
